# Multiband Microstrip Rectenna Using ZnO-Based Planar Schottky Diode for RF Energy Harvesting Applications

**DOI:** 10.3390/mi14051006

**Published:** 2023-05-06

**Authors:** Somaya I. Kayed, Dalia N. Elsheakh, Hesham A. Mohamed, Heba A. Shawkey

**Affiliations:** 1Obour High Institute for Engineering and Technology, Cairo 11828, Egypt; 2Department of Electrical Engineering, Faculty of Engineering and Technology, Badr University in Cairo, Badr City 11829, Egypt; 3Electronics Research Institute, Cairo 11843, Egypt; 4National Telecommunications Regulatory Authority, Giza 12577, Egypt

**Keywords:** microstrip antenna, rectenna circuit, moon-shaped cut, radio frequency (RF), energy harvesting (EH), planar Schottky diode (PSD), Ag/ZnO

## Abstract

This paper presents a single-substrate microstrip rectenna for dedicated radio frequency energy harvesting applications. The proposed configuration of the rectenna circuit is composed of a clipart moon-shaped cut in order to improve the antenna impedance bandwidth. The curvature of the ground plane is modified with a simple U-shaped slot etched into it to improve the antenna bandwidth by changing the current distribution; therefore, this affects the inductance and capacitance embedded into the ground plane. The linear polarized ultra-wide bandwidth (UWB) antenna is achieved by using 50 Ω microstrip line and build on Roger 3003 substrate with an area of 32 × 31 mm^2^. The operating bandwidth of the proposed UWB antenna extended from 3 GHz to 25 GHz at −6 dB reflection coefficient (VSWR ≤ 3) and extended from both 3.5 to 12 GHz, from 16 up to 22 GHz at −10 dB impedance bandwidth (VSWR ≤ 2). This was used to harvest RF energy from most of the wireless communication bands. In addition, the proposed antenna integrates with the rectifier circuit to create the rectenna system. Moreover, to implement the shunt half-wave rectifier (SHWR) circuit, a planar Ag/ZnO Schottky diode uses a diode area of 1 × 1 mm^2^. The proposed diode is investigated and designed, and its S-parameter is measured for use in the circuit rectifier design. The proposed rectifier has a total area of 40 × 9 mm^2^ and operates at different resonant frequencies, namely 3.5 GHz, 6 GHz, 8 GHz, 10 GHz and 18 GHz, with a good agreement between simulation and measurement. The maximum measured output DC voltage of the rectenna circuit is 600 mV with a maximum measured efficiency of 25% at 3.5 GHz, with an input power level of 0 dBm at a rectifier load of 300 Ω.

## 1. Introduction

The progress of portable electronic devices in this modern era depends mainly on wireless data transmission technology. This technology has evolved from designing an external antenna to an internal antenna in advanced electronic devices, which requires necessary changes in the shape and performance of the antenna [1]. These advanced electronic devices perform a variety of intelligent functions such as sensing, automation, health monitoring, energy harvesting, etc. In order to ensure uninterrupted data transmission and to take advantage of these features, wireless communication requires a large bandwidth [2]. This large band is UWB, as it has an unlicensed frequency band ranging from 3.1 to 10.6 GHz and is intended for commercial use [1]. The bandwidth of 7.5 GHz has had increased attention and led to great developments in the use of wireless communication systems by using shorter electromagnetic pulses. The planar microstrip antenna is in high demand for use due to its easy installation in the transceiver system. There have been efforts to design and develop different types of broadband antennas with stable radiation patterns, such as planar monopole, spiral antennas, log periodic and bowtie [1,2,3]. For UWB wireless communication applications, printed monopole antennas are very promising due to their advantages such as low cost, compactness, and wide bandwidth [2,3]. The first step in achieving UWB range is to create a standard rectangular or circular-shaped monopole antenna. By adjusting the radiator and ground-plane shape of the monopole antenna, it affects the transmission line characteristics and the distribution of the current. Various antenna configurations for achieving UWB impedance matching have been described in several articles published previously. In [4], an L-shaped inverted antenna with a 50 Ω transmission line feed and a square aperture is used. Geometric optimization of the feedline and slot achieves impedance bandwidths extending from 3.5 to 9.5 GHz. In [5], the combination of different geometries with a coaxial ladder slot, coplanar waveguide (CPW) ground structure with a UWB monopole antenna is demonstrated. In addition, in [6], the multi-input–multi-output (MIMO) configuration of the proposed antenna shows that ultra-wideband (UWB) 5G communications are best suited for an end-user customer to accept and reject different communication services simply by utilizing the corresponding band. The UWB–MIMO isolation and gain of the antenna are enhanced using a parasitic isolator, and the surface is also shown as a radiator in the form of an arc with a lower ground plane composed of parasitic components in [7,8], respectively. The UWB monopole antenna is given in [9,10], consisting of a radiator based on a spline optimizer and a defected ground structure (DGS) [11]. Due to waste portions of radiated energy in undesirable directions, it is very efficient to use broadband dual interaction antennas to achieve powerful detection and sensing systems [12,13,14,15,16,17,18]. It is very important that wireless communication applications such as Worldwide Interoperability for Microwave Access (WiMAX), wireless local-area network (WLAN), lower band of 5G, and short-range communications (SRC) can be integrated easily into smart vehicles, where antenna size and placement are vital [19]. Moreover, the unlicensed frequency range from 5.470 to 5.725 GHz benefits intelligent transportation systems (ITS), whereas the frequency range from 3.168 to 4.752 GHz in densely populated areas provides multi-path immunity to fast moving vehicles [20]. A multiband rectenna (antenna connected to a rectifier circuit) capable of harvesting ambient energy in different frequency bands will increase the use of battery-less portable electronics, implantable devices, and a wide range of IoT sensors. Many diode rectifier topologies exist to convert the received RF power into DC signals, whereas the matching circuit and Schottky diode device determine conversion efficiency. Multiple printed electronics techniques [21,22] allow for the simplified synthesis of planar Schottky diodes (PSDs) in PCB technology. Among the large number of metal oxide semiconductor materials, ZnO arises due to its wide bandgap, good chemical and thermal stability, and high mobility, with high operation frequencies that allow its usage for wideband applications [23,24,25,26,27,28,29].

In this paper, a complete multiband rectenna is designed and fabricated using an in situ synthesized Ag/ZnO planar Schottky diode. First, a low-profile linear-polarized UWB monopole harvest antenna was designed with the working frequency to cover most wireless communication, with a lower sub-band of 5G and UWB ranging from 2.5 GHz to 25 GHz. A metallic patch radiator in the form of a clipart moon-shaped cut with a semi-circular ground plane along with a U-shaped stub provides good impedance matching. The proposed UWB antenna has a bandwidth of 20 GHz and a peak gain of 5.1 dBi, with omnidirectional characteristics in most of the working bands. Two 3D electromagnetic software were designed and simulated the proposed antenna using computer simulation technology (CST) and Ansys high-frequency structure simulator (HFSS). The genetic algorithm optimization in HFSS version 15 is used to optimize the rectenna dimensions. A planar Schottky diode (PSD) consists of an n-type semiconductor ZnO with an Ag electrode and is synthesized using ADS 2008 (Advanced Design System free version), with both DC and RF performance used for the rectifier design. The synthesized PSD has a current density of 0.1 mA/cm^2^ and operates in wideband frequency range extending from 3 GHz to 10 GHz and from 16 GHz to 20 GHz. The shunt half-wave rectifier (SHWR) topology shown in Figure 1 is selected for rectenna implementation. The organization of this work as follows: Section 2 explains the antenna design, describes the antenna performance, and shows the antenna performance together with a comparison between the simulated and the experimental results. Section 3 describes the SHWR fabrication and characterization. Complete rectenna measurements are demonstrated in Section 3. Finally, Section 4 presents the conclusions.

## 2. Antenna Design

The design of the proposed monopole antenna is discussed in this section in terms of design procedure, current distribution, fabrication, measured results, and the radiation pattern.

### 2.1. Proposed UWB Antenna Design Approach

The proposed antenna is a compact printed antenna with overall dimensions of 32 × 31 × 1.575 mm^3^. The antenna could be easily integrated on the same substrate with a rectifier circuit to assist in the RF energy harvesting applications of the mobile device. Figure 2 shows the 3D geometry of the proposed monopole antenna, and it is designed and fabricated using Rogers RO-3003 substrate with a relative dielectric constant of *ε_r_* = 3, and thicknesses of *h* and *tan σ*, as shown in Table 1. The antenna is fed using a 50 Ω feeding line with a width of *W_o_* and a length of *L_o_*. The ground plane has a rectangular shape with an etched rectangular slot and a curved shape at the top. The antenna’s dimensions are carefully chosen and optimized using both Microwave Studio CST version 2022 and HFSS version 15, and the dimensions values of the designed antenna as marked in Figure 2 are shown in Table 1. The proposed antenna design underwent a number of steps to obtain the final design as shown in next section.

### 2.2. Design Procedure of the UWB Antenna

This section displays and discusses the design steps of the developed antenna. The detailed structure of the linear polarized UWB monopole antenna is shown in Figure 2. It could be observed that the antenna consists of a radiating patch shaped by the impedance matching of the feedline (*W_o_*, *L_o_*), which is 50 Ω, and two exclusive non-central circular patches with inner cuts for obtaining the required frequency band-shape slot on the top and a semicircular ground plane with a rectangular truncation on the bottom.

The effective dielectric constant ($\varepsilon_{reff}$) could be expressed as:(1)$$\varepsilon_{reff}=\left(\frac{\varepsilon_{r}+1}{2}\right)+\left(\frac{\varepsilon_{r}-1}{2}\right){\left(1+\frac{12\times h}{w}\right)}^{-\frac{1}{2}}$$
where *h*, *w* are the hight and width of the transmission line.

The resonance frequency of the clipart moon-shaped radiating patch could be calculated using Equations (2) and (3), according to the standard formula [8,9].

The first resonant frequency *f_r_*
(2)$$f_{r}=\frac{1.8411c}{2\pi D_{1}{\left[\varepsilon_{reff}\left\{1+\frac{2h}{\pi {L\varepsilon }_{r}}\left(ln\left(\frac{L}{2h}\right)+\left(1.44\varepsilon_{r}+1.77\right)+\frac{h}{L}\left(0.258\varepsilon_{r}+1.65\right)\right)\right\}\right]}^{1/2}}≅\frac{c}{4*L*\sqrt{\varepsilon_{reff}}}$$
(3)$$L=D_{1}+(L_{o}-(L_{2}-L_{1}))$$
where (c) is the velocity of light, ($f_{r}$) is the resonant of frequency, *D*_1_ is the outer dimension of the clipart moon shaped, ($\varepsilon_{r}$) is the dielectric constant of substrate, and *h* is the height of the substrate. Therefore, using the above equations; $\varepsilon_{reff}=2.9$, then the first resonant frequency is 3.6 GHz, the ground plane has curvature, and the simple clipart moon-shaped cut is used to improve the impedance bandwidth and reduce the resonant frequency to 3 GHz. The proposed UWB antenna is fed using the 50 Ω microstrip line. The curvature and simple rectangular U-shaped slot in the design change the current distribution and affect the inductance and capacitance; they are incorporated into the ground plane and help the antenna to exhibit a wide bandwidth. The advantage of our design is its simplicity, and also the integration between the planar structure and other microwave circuits. The design dimensions are given in Table 1. Figure 3 shows the different design stages until the proposed final structure and the reflection coefficient simulation results |S_11_| are reached. The design phases associated with the proposed antenna design procedure are shown in Figure 4.

Figure 3a shows the first step in the design of the proposed monopole. A circular radiator fed by a 50 Ω transmission line is employed as a reference monopole antenna (1). The circular radiator width *W_o_* and length *L_o_* are calculated with the help of the theory of transmission lines [27,28].

Figure 3a shows that the monopole antenna (1) has poor matching at shields as the bandwidth extends from 3 GHz to 22 GHz with many notches in the operating band, as shown in Figure 4.

The second step of the design antenna (2) is the etching of small circles, which has an important effect on the impedance bandwidth, as shown in Figure 3b. When *D*_2_ = 7.2 mm, it creates more intense resonances and generally decreases the return loss level and better matches than the antenna (1), as shown in Figure 4. Notice that the antenna (2) has two operating frequencies, 10.3 GHz and 14.2 GHz, with a bandwidth extended from 2.5 GHz to 20.3 GHz and notched from band 6 to 10 GHz, as well as from band 14 GHz to 16 GHz, as shown in Figure 4.

For the third design step, a U-shaped slot is etched under the 50 Ω feed transmission line, as shown in Figure 3c. This slot is used to provide an impedance bandwidth from 2.5 GHz to 22 GHz, with a notched extension from 5 to 9 GHz, as shown in Figure 4.

The final design step of the proposed monopole is achieved using a modified ground plane to increase the electrical length by etching a semi-circular defective ground structure in the ground plane. The current distribution is perturbed by inserting an elliptical slot in the radiator, which yields the creation of a novel wideband around 15 GHz and enhances the impedance bandwidth of the final shape to extend from 2.5 GHz to 22 GHz, as shown in Figure 4. Thus, this covers the entire UWB range, as shown in stage 4 of Figure 3d.

### 2.3. The Current Distribution of UWB Antenna

The simulated surface current distribution of the proposed antenna at different resonant frequencies is shown in Table 2. This table shows the resonance modes to visually emphasize the coupling effect of radiated element and ground plane geometry. The current distribution on the antenna is simulated as the current pattern at different resonance frequencies over the operational band at 3.5, 12.5, 14.5, 15.5, 18.5, and 22 GHz, depicting different frequencies which are used for good understanding of the proper excitation of the matching impedance bandwidth. According to the current distribution pattern, the radiating patch and ground plane’s edges are where the majority of currents are found, whereas these area centers have very weak currents. Additionally demonstrated is the current’s coupling to the patch through the 50 transmission feed lines from the top and bottom edges of the ground plane before radiating into free space. By changing the current flow and generating a symmetrical current distribution to the modified ground plane, the truncation on the top edge of the partial ground plane effectively reduces the fluctuations in antenna impedance and lessens the impact of the ground plane on the antenna’s performance. However, at higher frequencies, the currents are mainly distributed on the microstrip transmission line and the junction between the patch and ground plane. Thus, the currents on the ground plane become stronger than the lower frequencies, and the impedance matching becomes worse for traveling wave-dependent modes.

### 2.4. Measured Results of Proposed Antenna

The proposed monopole antenna is a fabricated and manufactured prototype of the proposed UWB antenna as shown in Figure 5a front part and Figure 5b back part. Furthermore, the antenna parameters such as the reflection coefficient |S_11_| and voltage standing wave ratio (VSWR) of the proposed monopole antenna versus frequency with optimal design are shown in Figure 5c,d. All measurements were performed using Rohde and Schwarz’s ZVA 67 Vector Network Analyzer. Figure 5c shows a comparison between the simulated and measured results of the reflection coefficient and have good agreement with each other. The simulated and the experimental results ensure that the antenna covers all the aforementioned mobile and wireless applications bands. Because of the amazing development in electronic circuits, the sensitivity and accuracy of signal extraction increased. Moreover, the characteristic bandwidth impedance at −6 dB reflection coefficient was referenced in the frequency band from 2.5 GHz to 20 GHz, which gives a better ultra-wideband performance. The simulated and measured VSWR at various operating frequencies can be determined from the VSWR plot shown in Figure 5d. The photo of the fabricated antenna with the vector network analyzer for proposed antenna reflection coefficient and voltage standing wave ratio are shown in Figure 5e,f, respectively. In general, it is noted from the previous results that there is a good agreement and good impedance matching obtained in the frequency range extending from 2.5 to 22 GHz. The differences between simulated and measured results at low and high frequencies could be attributed to several reasons: First, the effects of the coaxial cable used in the measurement. Second, the SMA solder used to solder the SMA conductor to the feeder line. Third, other electromagnetic interference signals in the atmosphere and ideal model uses for simulation, as well as manufacturing and measurement tolerances in the positioned antenna.

### 2.5. Far-Field Properties

In this section, the far-field radiation patterns and gain at some matched frequencies could be achieved. The photo of the measure radiation pattern setup is custom-made in the electronics Research Institute, Microstrip Lab, and shown in Figure 6. The simulated three-dimension radiation pattern of the proposed monopole antenna at the frequency is shown in Table 3. The comparison between measured and simulated radiation pattern obtained for the proposed antenna at 3.5 GHz, 12.5 GHz, 14.5 GHz, and 17.5 GHz in the X-Z plane and X-Y plane of the Phi = 0° (H-plane) and phi = 90° (E-plane), respectively, is shown in Table 3. From this table, it is clear that the simulation used by the CST simulator and from the measured results, shows that the directive pattern of radiation in the H-plane is near omnidirectional, and in the E-plane it is bidirectional and stable. Moreover, it is noted that the shape of the radiation for the low frequencies at 3.5 GHz is not symmetrical in the plane phi = 90° due to the shape of the radiation patch, which is not symmetrical with respect to the Y-axis. This configuration makes the antenna a little more directive on one plane compared with the other plane, which is the opposite to that shown in the radiation pattern. In addition, at high frequencies, the radiation pattern becomes omniradiational; because of the far field, the radiation pattern differs with the resonant frequency, according to the main element of antenna contributed to the radiation. According to the current distributed over the surface of the monopole, the radiation pattern is changed. There are differences between measured and simulated radiation patterns, especially at high frequencies. These could be attributed to the custom-made anechoic chamber, as shown in Figure 6 in the ERI Lab. Moreover, the reference homemade horn antenna it is not exact ultra wideband over all the operating frequencies. As well as the absorber material not being used for high frequencies, it is limited to 5 GHz.

The comparison between simulation and measurement of the gain and radiation efficiency of the proposed UWB antenna is shown in Figure 7a,b, respectively. The proposed antenna provides the simulated gain of 4.2, 5.1, 5.2, and 4.1 dBi at 12.5, 14.5, 15.5, and 17.5 GHz, respectively, where the proposed antenna has a high gain of 4.5 dBi and more than 80% efficiency overall. There is a difference between the two electromagnetic simulators used to design the proposed monopole antenna (HFSS and CST). These are due to several factors, the first of which is the type of numerical method used to analyze the structure. HFSS used the finite element method (FEM), whereas CST used the finite integration technique (FIT) and the transmission line matrix (TLM). Second, the dimension of the radiation pattern used in the design of the proposed monopole antenna in HFSS is fixed and not less than by λ/4 of the operating resonant frequency at a single frequency. In CST, it is automatically assigned from the simulator and adapted according to frequency. Then, the measured gain is calculated using the gain comparison method, where the received power of the antenna under test (AUT) is compared with a reference horn antenna (OBH-20200-C), and the bandwidth is extended from 2 to 20 GHz with a known gain 8.25 dBi. If different connectors are required to connect the two antennas, their attenuation has to be taken into account. Therefore, the difference in results could be attributed from the differences in matching conditions in the simulation scenario and the actual antenna. Moreover, antenna pattern and gain measurements are quite unlike most microwave component tests. This is because they need to be performed in an open-space environment with well-controlled minimum reflections, especially if a vector network analyzer (VNA) and spectrum analyzer are added for high-frequency wave bands. In addition, the interactions introduced by the test equipment and test fixtures and/or equipment holders and the isolation shield (foam absorbers material) are fixed on the anechoic chamber room wall.

The antenna radiation efficiency is measured over the operating bands using the Wheeler cap method [14,15] at the resonant frequencies as shown in Figure 7b. The average radiation efficiency is around 75% over the operating bands.

Table 4 presents the comparison between the current work and other related published papers using the design of the UWB monopole antenna. This table contains different types of UWB antennas operated at different applications. Table 4 shows that the proposed antenna with the smallest dimensions from other published works have achieved acceptable gain and efficiency.

## 3. Shunt Half-Wave Rectifier (SHWR)

The Schottky diode consists of one layer of a p-type or n-type semiconductor positioned between two metal sides: one represents the electrode which allows charge injection in one direction, while it is blocked in the other [21]. Among the semiconductor devices used to implement thin film scotch diodes, ZnO is a promising candidate for use in printed diodes, providing a potentially cost-effective and scalable solution for a range of electronic applications. Moreover, its high breakdown voltage (>15 V) and high rectification ratio compared with other diodes make this environmentally friendly process a serious option for power electronics and energy harvesting [24]. In this paper, ZnO n-type semiconductor with Ag electrode is used to implement PSD for multiband operation with total dimensions 1 × 1 mm^2^. The proposed diode consists of Ag/ZnO layers supported on a copper layer with the scotch barrier at the Ag/ZnO junction. In this paper, a ZnO n-type semiconductor with an Ag electrode is used to implement PSD for multiband operation with total dimensions of 1 × 1 mm^2^.

### 3.1. Ag/ZnO PSD Fabrication and Characterization

The PSD is first fabricated on the same substrate as Rogers (RO3003) and characterized using the design of the rectifier circuit. The device is implemented using nano electronics fabrication, similar to ZnO n-type semiconductor nanoparticle ink (Sigma-Aldrich, Darmstadt, Germany) and commercial silver paste Ag electrode with size 1 × 1 mm^2^. A wideband transmission line (WBTL) is first designed using the ratio of the width to length of the WBTL equal to 3 mm/20 mm with 0.25 mm separation and it is implemented on Rogers 3003 substrate. The measured WBTL reflection coefficient is less than −20 dBm in the frequency range from 2 GHz to 25 GHz; the proposed diode is then applied on it to investigate the performance of the fabricated diode.

Figure 8a shows the PSD structure, consisting of a thin layer of ZnO on the WBTL and a layer of Ag. The ZnO layer is deposited by dropping the nanoparticle ink on one side of the WBTL (using sonoplot GIX microplotter II) and heating the layer to 100 °C; this process is repeated 10 times to achieve a uniform film with 200 nm thickness (measured using the FX20 thin film analyzer). The Ag layer is then dropped manually onto the ZnO layer, dried to 150 °C, and then connected to the other side of the WBTL using a commercial flexible copper tape (35 μm thickness). Figure 8b shows a photo of the PSD deposited on the copper transmission line; before the commercial flexible copper tape is added, only the upper Ag layer appears using the digital microscope camera 50×. The device |S_11_| measured with one side of the WBTL is connected to the ground to obtain the device S-parameter shunt connection performance. To investigate the PSD diode performance, its I/V characteristics are measured using the Keithely 4200 SCS (semiconductor characterization system), as shown in Figure 8b. The proposed PSD has a total area of 1 × 1 mm^2^, and a maximum current density 0.1 mA/cm^2^ with an I_On_/I_Off_ current ratio equal to four. This is due to the usage of commercial grades of Ag and copper tape; applying a scientific-grade material can improve diode performance. The S-parameter performance of the diode is shown in Figure 8c, with S-parameters resonance frequency extending from 3 GHz to 10 GHz and from 16 GHz to 20 GHz, according to IEEE standardization.

### 3.2. Rectifier Design and Implementation

A single matching circuit is designed for multiband operation; this reduces the efficiency compared with the single-frequency matching circuit, but shows a good impact of reduction on the rectifier area. Figure 9a shows the design of SHWR—shown in Figure 1—where TL1 represents the matching network while TL5/C2/TL6 represents the DC pass filter using the ADS simulator. The fabricated planar Schottky diodes (PSD) are shown in Figure 9b using the photolithographic technique. The PSD-measured S-parameters (Figure 8c) are used for simulation, whereas TL3 and TL4 connected to the PSD have the same width (3 mm) of wideband TLs used for PSD characterization, as shown in Figure 9c. For multiband operation, the rectifier circuit is designed by applying a parametric sweep to all TLs dimensions to obtain wideband operation. Figure 9c shows the simulated rectifier without a diode and the comparison between the simulation and measurements of the complete rectifier circuit, where the difference could be explained due to the effect of the DC pass filter load on the rectifier, whereas the S-parameter of the diode (used for rectifier design) is measured considering only the effects of TL3 and TL4. The measured results show that the proposed rectifier operates at different resonant frequencies, at 3.5 GHz, 6 GHz, 8 GHz, 10 GHz and 18 GHz. This operating bands show a good performance for the sub band of 5G, with lower band applications of 3.5 GHz and 6 GHz, and UWB (3.5–10 GHz) and different high frequencies. The total rectifier circuit area is 40 × 9 mm^2^, as shown in Figure 9a, and the other transmission lines dimensions are shown in Table 5, where, *C*_1_ = *C*_2_ = 2*pF*, whereas the resistive load could be changed with no effect on the S-parameter.

To determine the optimum load used of the rectifier operation, simulations for RF-DC conversion are carried out at different load resistances; then, the RF input signal is applied to the rectifier at different operation frequencies of 3.5 GHz, 6 GHz, 10 GHz, and 18 GHz. The output DC voltage and conversion efficiency simulations for three resistive load values, 300 Ω, 1 kΩ, and 5 kΩ, are shown in Figure 10a–c. From previous simulations, the 300 Ω load was selected for rectifier fabrication as it has the largest DC output. The rectifier has a considerable DC output voltage that can reach 750 mV, which is sufficient for biasing tiny sensors and portable electronics devices.

### 3.3. Rectenna Measurements

A load of 300 Ω is used for rectifier implementation, and the rectifier is connected to the antenna and complete rectenna DC output voltage *V_dc_* is measured. Figure 11a shows the measurements’ setup block diagram, where the rectenna is set 15 cm away from a fabricated antenna lab. The horn antenna is used as the transmitter that radiates the electromagnetic signal to the proposed rectenna, as shown in Figure 11b. A horn antenna with a maximum operating frequency of 10 GHz is connected to Anritsu RF source G3697C and the rectifier DC output *V_dc_* is measured and then used in Equations (4) and (5) to calculate the DC output power *P_DC_* and efficiency *η* as:(4)$$P_{DC}=\frac{V_{dc}^{2}}{R_{load}}$$
(5)$$\eta =\frac{P_{DC}}{P_{in}}\%$$
where *R_load_* = 300 Ω and *P_in_* represents the RF input power.

An input power (*P_in_*) sweep from −10 dBm to 10 dBm is applied to the horn and electromagnetic power is radiated to antenna. Output DC from the rectifier is measured using an AVO meter. Figure 11c,d show the DC output volt with power and efficiency, respectively. The discrepancy between the simulation and measurement results could be explained due to measurement non-idealities such as distance between the transmitter horn and rectenna, misalignment, and other circumstances in the lab. The low efficiency could be explained due to the usage of commercial materials for diode implementation—Ag electrode and copper tape are used to connect the Ag electrode to the copper transmission lines. Using higher-grade materials can increase the efficiency. Moreover, the Ag electrode drops manually on the ZnO surface; applying advanced deposition techniques such as thermal evaporation (which is not available for authors) can improve diode performance. For input power levels of 0 dBm, the proposed rectenna has output DC voltages of 250 mV and 0.21 mW DC, which are able to bias the modern low power electronic modules and tiny sensors.

Table 6 shows the comparison between the proposed rectenna and other previously published architectures with the in situ implementation of Schottky diodes. Comparison shows that the proposed architecture has multiple resonance frequencies that enable the RF to harvest energy at different bands of operation, with the highest frequency at 18 GHz, compared with other architectures. The proposed rectenna has a reasonable DC output voltage at different loads, makes it a good candidate for energy harvesting applications. The moderate efficiency and measured DC output voltage are due to the distance between the transmitter and rectenna (about 15 cm) determined by the measurement setup compared with 2.5 cm for Ref. [13]. The measured values are suitable for portable electronics and IoT applications.

## 4. Conclusions

This paper presented the design and implementation of multiband rectenna for RF energy harvesting applications. An UWB moon-shaped cut-printed monopole antenna was designed with a defected curvature ground plane and U-shaped slot to improve the antenna bandwidth by changing the inductance and capacitance in the ground plane. The proposed UWB antenna is fed by a 50 Ω transmission line and built on RO 3003 substrate with an area of 32 × 31 mm^2^. A shunt half-wave rectifier with an Ag/ZnO PSD is proposed. The PSD has a current density of 0.1 mA/cm^2^ and has a wideband frequency extending from 3 to 10 GHz and from 16 to 20 GHz, with an area of 1 × 1 mm^2^; the measured S-parameters of the PSD are used to design the SHWR rectifier. The proposed multiband rectifier operates at 3.5 GHz, 6 GHz, 8 GHz, 10 GHz, and 18 GHz and has an O/P DC voltage of 400 mV at 0 dBm I/P RF signal, with a total area of 40 × 9 mm^2^. The complete rectenna has an O/P DC voltage of 250 mV and 0.21 mW DC power at zero dBm received signal, which can be used for powered portable electronics and tiny sensors with a maximum efficiency of 25% at 3.5 GHz. Aside from its simple fabrication technique and multiband of operation, it is a good choice for RF energy-harvesting modules at different wireless communication applications for battery-less electronics applications.

## Figures and Tables

**Figure 1 micromachines-14-01006-f001:**
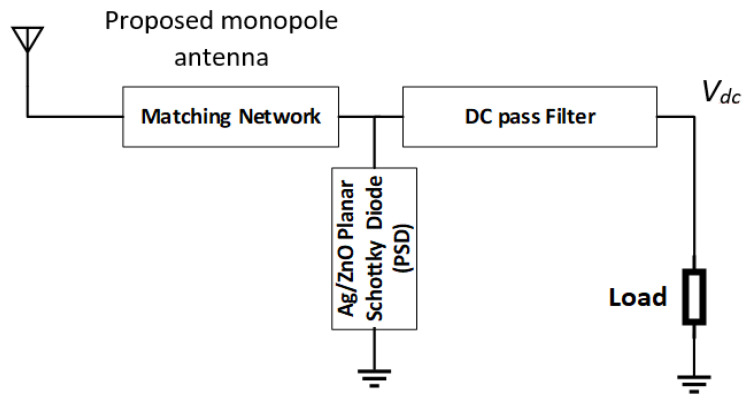
Proposed rectenna with shunt half-wave rectifier block diagram.

**Figure 2 micromachines-14-01006-f002:**
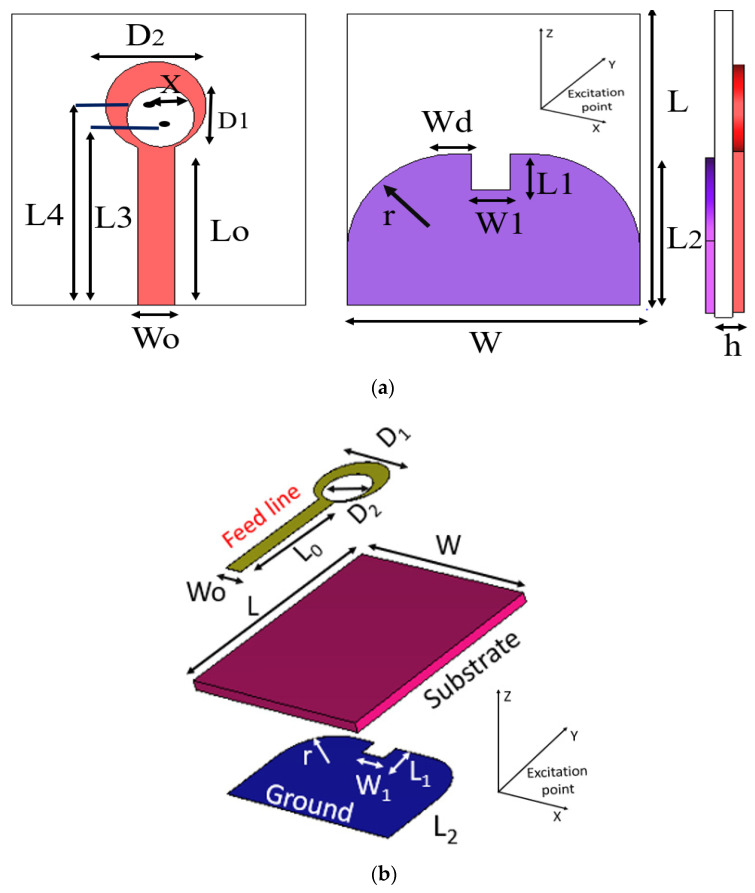
Proposed monopole antenna: (**a**) 2D geometry and (**b**) 3D geometry.

**Figure 3 micromachines-14-01006-f003:**
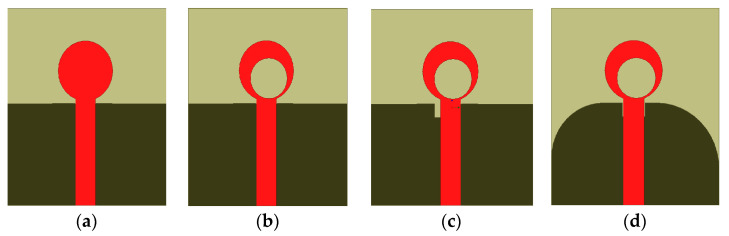
(**a**–**d**) Design procedure of the developed proposed UWB antenna. (**a**) Antenna (1); (**b**) Antenna (2); (**c**) Antenna (3); (**d**) Antenna (4).

**Figure 4 micromachines-14-01006-f004:**
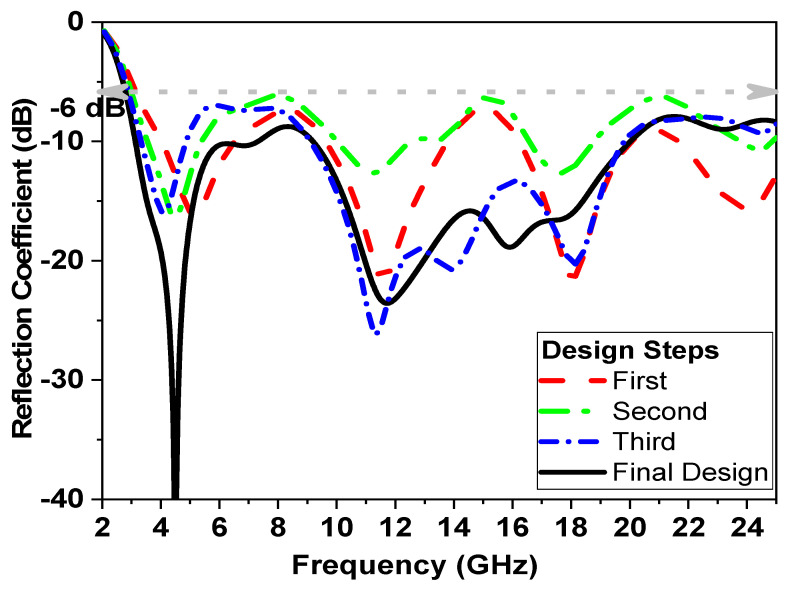
Simulated reflection coefficient for several proposed monopole design steps.

**Figure 5 micromachines-14-01006-f005:**
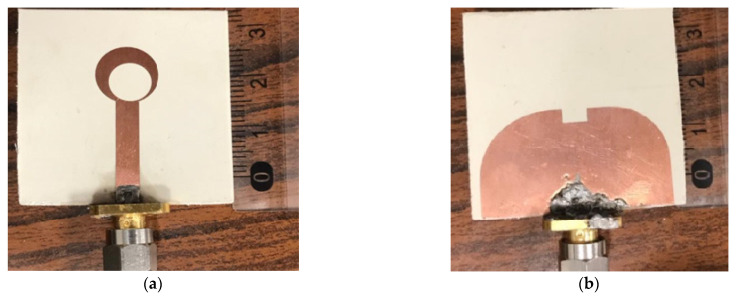
Manufactured prototype of the developed UWB antenna: (**a**) front part, (**b**) back part, the Simulated and measured of the proposed antenna of the UWB antenna structure (**c**) |S11| and (**d**) VSWR.

**Figure 6 micromachines-14-01006-f006:**
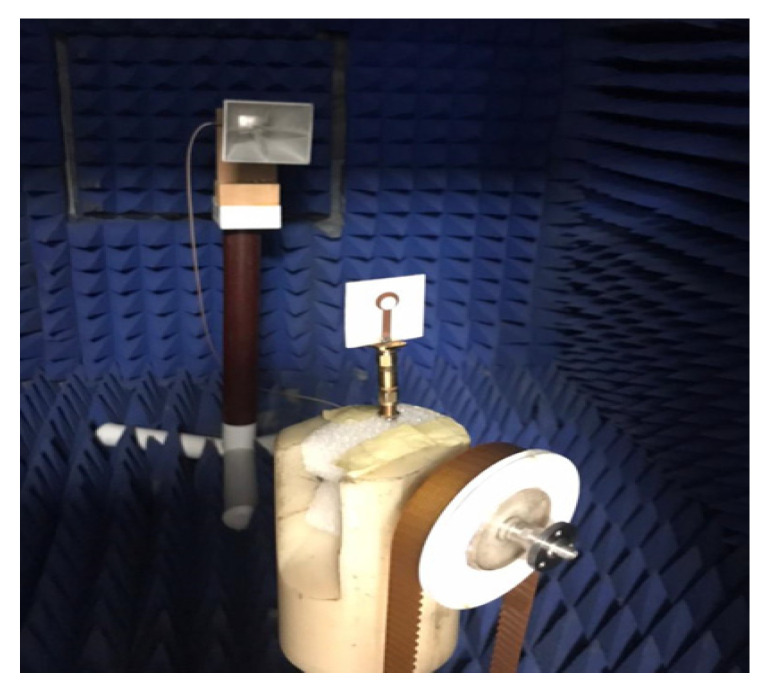
Measurement setup of the radiation pattern and peak gain.

**Figure 7 micromachines-14-01006-f007:**
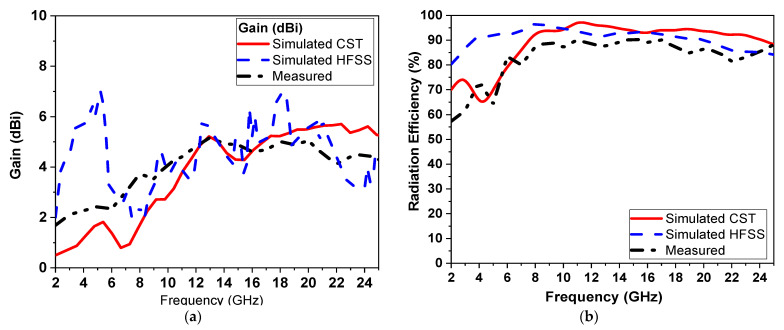
Simulated and measured (**a**) peak gain and (**b**) radiation efficiency versus frequency.

**Figure 8 micromachines-14-01006-f008:**
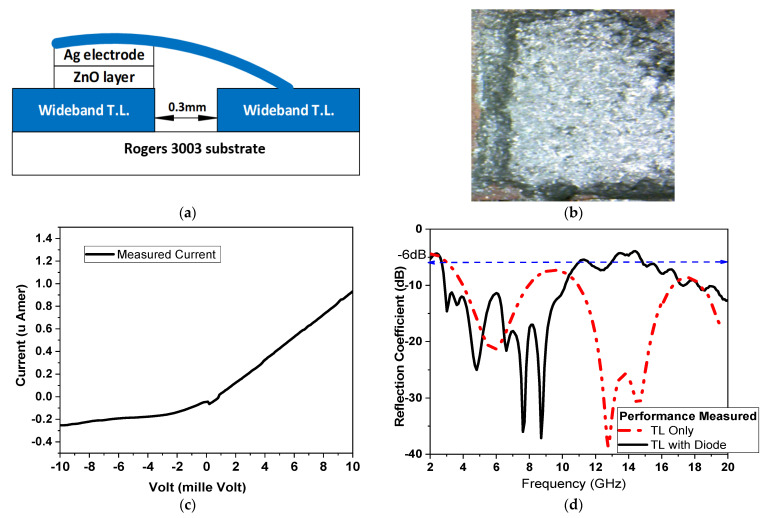
Ag/ZnO PSD. (**a**) Structure, (**b**) photo of PSD, (**c**) measured I/V characteristics and (**d**) measured S-parameter.

**Figure 9 micromachines-14-01006-f009:**
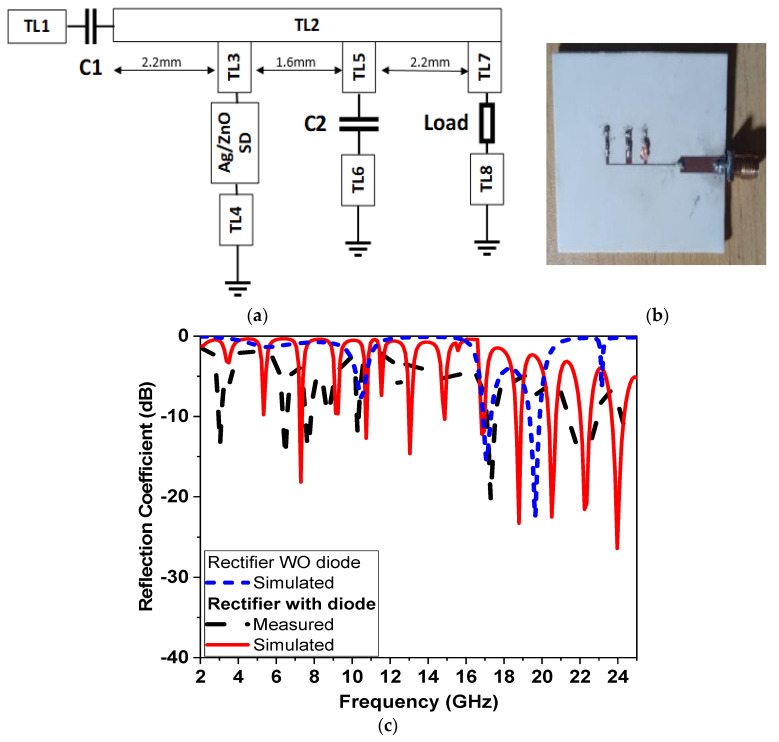
Shunt half-wave rectifier. (**a**) Circuit diagram. (**b**) Photo of the fabricated (**c**) S-parameter.

**Figure 10 micromachines-14-01006-f010:**
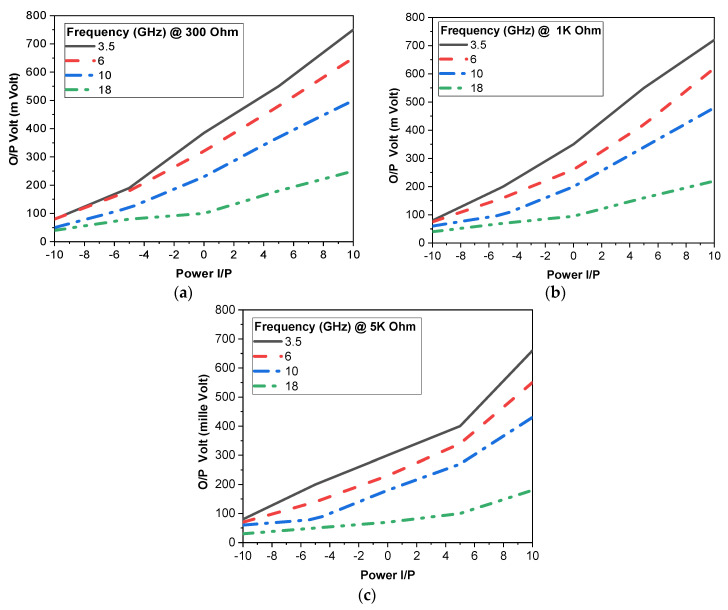
PSD simulated output DC Voltage.

**Figure 11 micromachines-14-01006-f011:**
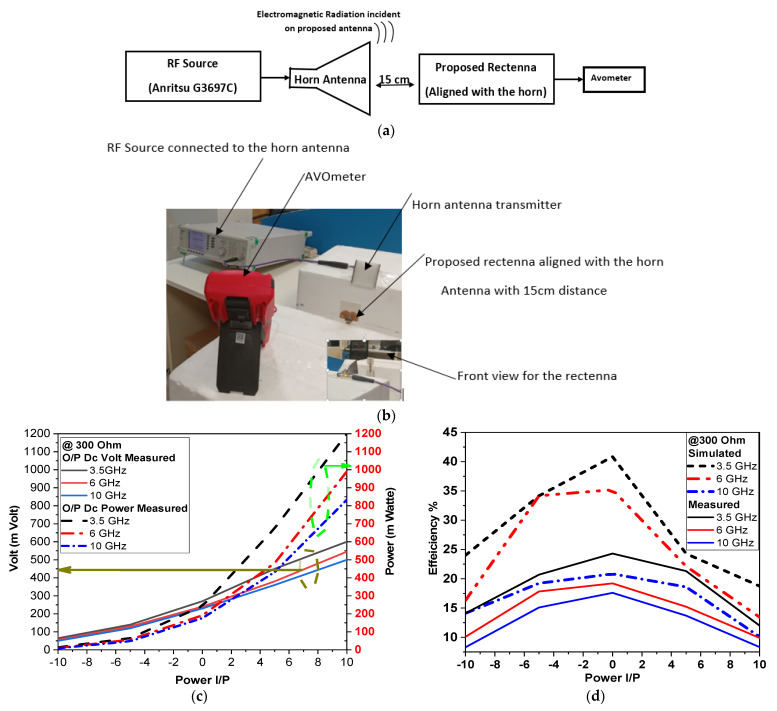
The setup of the RF energy harvesting: (**a**) configuration, (**b**) measurements of the proposed system, (**c**) output volt and power at 300 ohm and (**d**) comparison of simulated and measured efficiency.

**Table 1 micromachines-14-01006-t001:** Dimensions of the proposed antenna (all dimensions in mm).

*W*	*L*	*W_o_*	*L_o_*	*h*	*W* _1_	*W_d_*	*L* _1_	*L* _2_	*L* _3_	*L* _4_	*X*	*r*	*D* _1_	*D* _2_	*tan σ*
31	32	3	19.5	1.575	4.3	2.35	18.7	18.7	23.3	24.6	1.4	11	11	7.4	0.0026

**Table 2 micromachines-14-01006-t002:** Current distribution of the antenna at different frequencies.

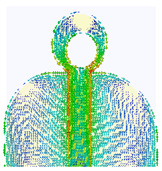	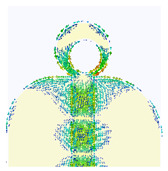	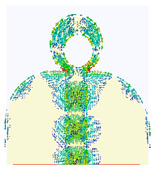
**3.5 GHz**	**12.5 GHz**	**14.5 GHz**
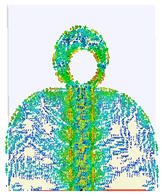	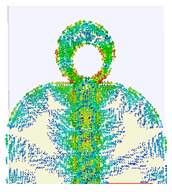	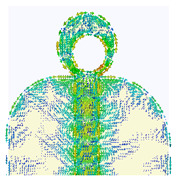
**15.5 GHz**	**18.5 GHz**	**22 GHz**

**Table 3 micromachines-14-01006-t003:** Far-field characteristics of proposed UWB antenna at different frequencies in the XZ and XY planes (simulated (solid line)—measured (dash line) with the simulated 3D radiation pattern).

*F* (GHz)	Phi = 0	3D Radiation	Phi = 90	3D Radiation
3.5	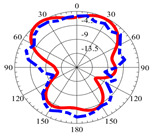	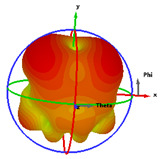	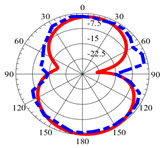	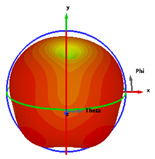
12.5	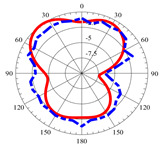	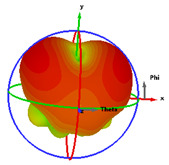	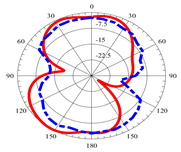	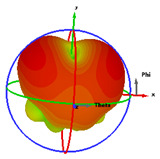
14.5	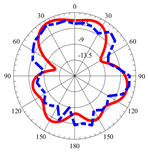	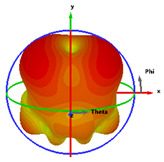	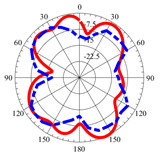	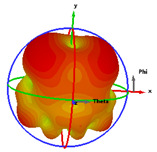
17.5	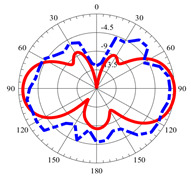	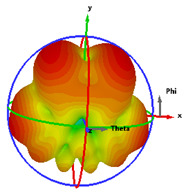	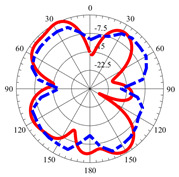	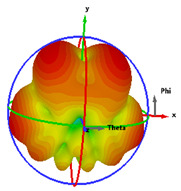

**Table 4 micromachines-14-01006-t004:** A comparison between the proposed antenna and other published papers.

Ref.	Size (mm^3^)	Size (λ_g_)^2^	Substrate	Res. Fre./BW	No. of Bands	|S11|(dB)	Application
[16]	45 × 57 × 1.5	2 × 1.24	FR4 ${є}_{r}$ = 4.4	1–13 GHz	1	−17	UWB
[17]	40 × 40 × 1.6	1.4 × 0.94	RT ${є}_{r}$ = 2.2	1.22–47.5 GHz	1	−14	UWB and 5G
[18]	62 × 64 × 1.6	1.26 × 0.86	FR4 ${є}_{r}$ = 4.4	1.68–26 GHz	1	−19	UWB
[19]	40 × 40 × 1.6	1.1 × 0.83	RT ${є}_{r}$ = 2.2	13.67, 15.28 GHz	2	−19	Satellite
[20]	50 × 50 × 1.6	0.66 × 0.66	FR4 ${є}_{r}$ = 4.4	25 GHz	1	−21	5G
This Work	32 × 31 × 1.575	0.31 × 0.21	RT4003 ${є}_{r}$ = 2.2	2.5 to 25 GHz	2	−25	5G and IoT

**Table 5 micromachines-14-01006-t005:** Rectifier circuit dimensions (all dimensions are in mm).

TL1	TL2	TL3	TL4	TL5	TL6	TL7	TL8
5/15	0.5/12	3/3	3/2	2/5	2/2	1/3	1/2

**Table 6 micromachines-14-01006-t006:** Performance comparison between proposed rectenna and previously reported architectures.

Ref.	Diode Structure	Antenna Structure	Max. Effic.	DC Output Volt	Frequency (GHz)
[25]	MoS2 Schottky barrier	Flexible substrate	40.1%	0.9 V/10 GHz, 0.5 V/15 GHz (p_in_ = 0 dBm)	2.4, 5.9, 10 and 15
[30]	Ni-NiO-Cr metal–insulator–metal (MIM)	microstrip slot	N/A	56 mV (p_in_ = 0 dBm)	2.45
[31]	Au/HfO2/Pt (MIM)	Bow Tie	N/A	0.25 mV (at p_in_ −20 dBm)	55–65
[32]	Graphene	Patch array	N/A	95 mV	28
[33]	MoS2 Schottky barrier	Commercial antenna	N/A	10 mV/2.4 GHz (p_in_ = 0 dBm)	0.1–10
[34]	SMS7630	Meander monopole	40%	N/A	2.4
[35]	SMS7630 and MA4E-1319	Broadband antipodal Vivaldi	12%	6.5 V (p_in_ = 20 dBm)	22.6–40
[36]	SMS7630	Dual ring-shaped monopole	40%	41 mV (p_in_ = 0 dBm)	1.8–2.6
[37]	HSMS 2860	Dual linearly polarized antenna array	40%	500 mV	1.8–2.1
This work	ZnO Schottky barrier	Moon-shaped cut	25%	250 mV/3.5 GHz (p_in_ = 0 dBm)	3.5, 6, 8, 10, 18

## Data Availability

Not applicable.

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
