# Peer review of "Multiband Microstrip Rectenna Using ZnO-Based Planar Schottky Diode for RF Energy Harvesting Applications"

_micromachines, 2023, doi:10.3390/mi14051006_

Round 1
Reviewer 1 Report (New Reviewer)
This work reported a Multiband Microstrip Rectenna Using ZnO Based Planar Schottky Diode for RF Energy Harvesting Applications. An UWB antenna operating in 2.5 – 25 GHz range with a multi-band rectifier design is studied.
The observations are as follows:
1. No novelty in the UWB antenna design. It is simply a ring shaped structure.
2. In general, -10dB level is considered for mentioning impedance bandwidth. The authors may change the mentions accordingly.
3. The measurement results (Fig. 5(e)) of the UWB antenna depicts that the VNA is not calibrated properly.
4. A clear study is needed to discuss why the proposed rectifier is presenting multi-band operation. Accordingly, mathematical analysis is also needed.
5. Moreover, no novelty in the structure. No new method or technique is reported/studied.
6. Huge discrepancy in rectifier simulation and measurement results. It needs to be justified or rectified.
7. Literature on rectifier circuits is missing. It needs to be added.
Author Response
Response Letter
Manuscript ID: micromachines-2296809
Manuscript Title: Multiband Rectenna Using ZnO Based Planar Schottky Diode for RF Energy Harvesting Applications
Type of manuscript: Article
Authors: Somaya Kayed, Dalia M. Elsheakh*, Hesham A Abdel Hady, Heba A. Shawkey
Dear Editor and Reviewers:
First, we would like to sincerely thank the editor and the reviewers for their valuable comments. We have prepared detailed responses for reviewer’ comments. In the revised paper, we use highlighted yellow text for modified sections in the paper based on the reviewer’ comments.
Sincerely yours,
Comments of Editor Comments:
This work reported a Multiband Microstrip Rectenna Using ZnO Based Planar Schottky Diode for RF Energy Harvesting Applications. An UWB antenna operating in 2.5 – 25 GHz range with a multi-band rectifier design is studied.
Comment 1: No novelty in the UWB antenna design. It is simply a ring shaped structure.
Response: Firstly, I would like to thank you for your kind attention to my article. Secondly, indeed, this is correct, using the monopole antenna is not a novel. The novelty of the given paper have highlighted in Introduction, page 1 and page 2 as follows. The shape and performance of this monopole is not published before, the antenna is composed of circular monopole antenna loaded with circular slot offset center and etched U shaped on the modified ground plane with overall compact size of 32×31 mm2. Performance of monopole antenna allows integration into system unlike similar work in literature. This configuration is used to improve the operational antenna bandwidth to extend from 3 GHz up to 25 GHz at -6 dB impedance bandwidth (VSWR ≤ 3). Moreover, when we used other scale at the -10 dB impedance bandwidth (VSWR ≤ 2) the operational antenna bandwidth to extend from 3.5 GHz up to 12 GHz and from 16 GHz up to 22 GHz. Moreover, design of multiband of rectifier circuit to harvest at multi frequency of operation. In addition, using on sit Ag/ZnO Schottky diode to complete the rectifier circuit.
Comment 2: In general, -10dB level is considered for mentioning impedance bandwidth. The authors may change the mentions accordingly.
Response: The proposed monopole antenna has reflection coefficient at -6 dB according to IEEE standard (VSWR≤3) (due to the huge revolution in the electronic circuit sensitivity) extend from 3 GHz up to 24 GHz. While, if we used -10 dB (VSWR ≤ 2) the characteristic impedance bandwidth of the proposed monopole antenna start to resonant at 3.5 GHz to 12 GHz and from 16GHz up to 22 GHz is accepted.
Comment 3: The measurement results (Fig. 5(e)) of the UWB antenna depicts that the VNA is not calibrated properly.
Response: We recalibrated The VNA and measured again the two parameters and saving them on the same time as shown in the following figure.
(a
- (b) (c)
Figure 1. (a) Vector network analyser (VNA) after calibration, (b) Reflection coefficient of the proposed antenna (c) VSWR of the proposed antenna.
(a) (b)
Figure 2. The Simulated and measured of the proposed antenna of the proposed structure (a) |S11| and (b) VSWR.
Comment 4: A clear study is needed to discuss why the proposed rectifier is presenting multi-band operation. Accordingly, mathematical analysis is also needed.
Response: A nanomaterial –used for diode implementation- can exhibit multiple frequencies in their electrical response due to several factors, including the presence of defects and impurities during manual deposition of the material in a non-clean environment. All these factors affect the bandgap which can give rise to multiple resonant modes. Due to all previous factors and the nonlinear performance of the diode in general, it is hard to obtain a mathematical model which will be inaccurate. To overcome the theatrical study, the diode frequency response is measured (Fig.8) and used for design and simulation of the rectifier.
Comment 5: Moreover, no novelty in the structure. No new method or technique is reported/studied.
Response: No novelty in diode implementation, the contribution is for using simple deposition technique to obtain a multiband rectifier.
Comment 6: Huge discrepancy in rectifier simulation and measurement results. It needs to be justified or rectified.
Response: Diode is first measured with simple wideband transmission line, the effect of other rectifier components wasn’t included, this can explain the discrepancy.
Comment 7: Literature on rectifier circuits is missing. It needs to be added.
Response: Added 3 references for rectifier circuit topology
- Almorabeti, S.; Rifi, M; and Terchoune,H.; “Rectifier Circuit Designs for RF Energy Harvesting applications”, HAL openscience, Sept.2019
- Zhou, Y.; Froppier,B.; and Razban, T.;“Schottky Diode Rectifier for Power Harvesting Application,” IEEE International Conference on RFID-Technologies and Applications (RFID-TA) Nov. 2012.
- Partal , H; Belen , M; and Partal, S; “Design and realization of an ultra-low power sensing RF energy harvesting module with its RF and DC sub-components,” International Journal of RF and Microwave Computer‐Aided Engineering, January 2019.

Reviewer 2 Report (New Reviewer)
"Multiband Microstrip Rectenna Using ZnO Based Planar Schottky Diode for RF Energy Harvesting Applications" presents an interesting work on a rectenna for RF energy harvesting. The self-made ZnO based planar diode is attractive, although some details should be discussed.
Here's my comments:
1. Fig. 5 (c) and (d) are the same. One figure is enough. Fig. 5 (e) and (f) are redundant as well. One is enough. Meanwhile, the printed screen should be deleted as well.
2. The explanation on the difference between Fig. 7 (a) and (b) are not reasonable. Please explain the CST simulation gain is lower than the measured gain.
3. Please show the measurement method of radiation efficiency in Fig. 7(b).
4. The Ag/ZnO PSD fabrication is interesting. How much volume of ZnO and Ag ink are used? And how to control the thickness of the ZnO layer to be 200nm?
5. A photo of the self-made diode, maybe under a microscope, should be presented.
6. In Fig. 9 (a) and (b), a shunt diode rectifier is presented. It seems that the photo and the layout are not consistent. The distance between C1 and TL3 is marked as 2.2mm. But in the photo, it is much greater than that.
7. Fig. 9 (c) shows the simulated results. I am wondering that a diode model should be presented before obtaining simulation results. Thus, please present the diode model used in the ADS at first.
Suggested revisions:
1. Equation (1) - (3) are not aligned well. Please correct them.
2. The first line after (3), "D1" should be "D_1", where 1 should be subscript.
3. "The proposed PSD has total area 1×1mm2 has a max.," has two "has". Please correct.
4. Equation (4) and (5)should be centered.
5. In the abstract, "40x9 mm2" should be "40x9 mm^2".
Author Response
Response Letter
Manuscript ID: micromachines-2296809
Manuscript Title: Multiband Rectenna Using ZnO Based Planar Schottky Diode for RF Energy Harvesting Applications
Type of manuscript: Article
Authors: Somaya Kayed, Dalia M. Elsheakh*, Hesham A Abdel Hady, Heba A. Shawkey
Dear Editor and Reviewers:
First, we would like to sincerely thank the editor and the reviewers for their valuable comments. We have prepared detailed responses for reviewer’ comments. In the revised paper, we use highlighted yellow text for modified sections in the paper based on the reviewer’ comments.
Sincerely yours,
Comments of Editor Comments:
This work reported a Multiband Microstrip Rectenna Using ZnO Based Planar Schottky Diode for RF Energy Harvesting Applications. An UWB antenna operating in 2.5 – 25 GHz range with a multi-band rectifier design is studied.
Comment 1: Fig. 5 (c) and (d) are the same. One figure is enough. Fig. 5 (e) and (f) are redundant as well. One is enough. Meanwhile, the printed screen should be deleted as well.
Response: Thank you very much for your effort in reviewing our paper. We added these figures according to the previous reviewers but according to your comment, we remove them from figure 5 as follow:
- Front (b) Back
(c) (d)
Figure 5. Manufactured prototype of the developed UWB antenna (a) front part, (b) back part, the Simulated and measured of the proposed antenna of the UWB antenna structure (c) |S11|and (d) VSWR.
Comment 2: The explanation on the difference between Fig. 7 (a) and (b) are not reasonable. Please explain the CST simulation gain is lower than the measured gain.
Response: The two simulations CST and HFSS of the antenna are performed in time and frequency domain respectively therefore, there is difference in gain results between two simulators. These simulators cannot take into account all aspects, including the material and finishing properties.
Then the measured gain is calculated by using the gain comparison method, where the received power of the antenna under test (AUT) is compared to a reference horn antenna (OBH-20200-C) extend bandwidth 2 to 20 GHz from of known gain 8.25 dBi. If different connectors are required to connect the two antennas, their attenuation has to be taken into account. The gain could be calculated from following Eq.
.
So, the difference in results could be attributed from the differences in matching condition in the simulation scenario and the actual antenna. Simulations have been performed considering a discrete excitation port of the antenna with definitions depending on the numerical tool. Measurements are referred to 50 ohm high precision connector and the characteristics of the isolation shield (foam absorbers material) fixed on the anechoic chamber room wall.
Moreover, antenna pattern and gain measurements are quite unlike most microwave component testing. This is because it needs to be performed in an open space environment with well-controlled minimum reflections especially if a Vector Network Analyzer (VNA) and Spectrum Analyzer are added for high frequency wave bands. In addition, the interactions introduced by the test equipment and test fixtures and/or equipment holders.
Any misalignment between the AUT and Transmitter Antenna or reflection caused by the text fixtures or any part of the testing environment can introduce measurement errors and uncertainties in the data.
- (b)
Figure 7. Simulated and measured (a) peak gain and (b) radiation efficiency versus frequency.
Comment 3: Please show the measurement method of radiation efficiency in Fig. 7(b).
Response: While the antenna radiation efficiency is measured over the operating bands using the Wheeler cap method [14, 15] at the resonant frequencies as shown in Figure 7(b). The average radiation efficiency is around 75% over the operating bands.
- P. Raiva and J.F. Sanchez, A rectangular cavity for cell phoneantenna efficiency measurement, In: IEEE International Symposiumon Antenna and Propagation, Washington, DC, 2005.
- M. Pozar and B. Kaufman, Comparison of three methods for themeasurement of printed antenna efficiency, IEEE Trans AntennasPropag., 36 (1988), 136–139.
Comment 4: The Ag/ZnO PSD fabrication is interesting. How much volume of ZnO and Ag ink are used? And how to control the thickness of the ZnO layer to be 200nm?
Response: As deposition is done manually, thickness cannot be controlled. Deposition conditions are estimated according to previous experience of microplotter users and the thickness is measured finally. Volume of Ag ink is 15µL (micro-liter) and of ZnO ink is 50 µL
Comment 5: A photo of the self-made diode, maybe under a microscope, should be presented.
Response: The photo of the self-made diode (PSD) is added to the update manuscript version (Fig.8(b)) by using digital microscope camera 50X and a paragraphs is added to explain this as follow:
“Figure 8 (b) shows a photo of the PSD deposited on the copper transmission line -before the commercial flexible copper tape is added- were only the upper Ag layer appears.”
- (b)
- (d)
Figure 8. Ag/ZnO PSD (a) structure, (b) photo of PSD, (c) Measured I/V characteristics and (d) Measured S-parameter.
Comment 6: In Fig. 9 (a) and (b), a shunt diode rectifier is presented. It seems that the photo and the layout are not consistent. The distance between C1 and TL3 is marked as 2.2mm. But in the photo, it is much greater than that.
Response: This can be considered due to that the captured image is extended in only 1 direction, rescaling the imaging is done.
- (b)
(c)
Figure 9. Shunt half-wave rectifier (a) Circuit diagram, (b) Photo of fabricated and (c) S-parameter.
Comment 7: Fig. 9 (c) shows the simulated results. I am wondering that a diode model should be presented before obtaining simulation results. Thus, please present the diode model used in the ADS at first.
Response: The model used for diode simulation is the S-parameter measured (by the VNA) for the diode before rectifier design (Fig.8(c), TL with load). The measured S-parameter is inserted to the simulation tool as a text file.
Comment 8:
- Equation (1) - (3) are not aligned well. Please correct them.
- The first line after (3), "D1" should be "D_1", where 1 should be subscript.
- "The proposed PSD has total area 1×1mm2 has a max.," has two "has". Please correct.
- Equation (4) and (5)should be centered.
- In the abstract, "40x9 mm2" should be "40x9 mm^2".
Response: All of these comments have been considered in the modified manuscript version.

Reviewer 3 Report (New Reviewer)
1. The abstract can be more concise, and the content does not need to be written using the software name, and can be placed in the main text. Units are not superscripted correctly.
2. The author designed the load to be 300 ohms, can it have better performance on other loads?
3. Figure 11 can provide a system architecture diagram, and restate enough details and definitions in Figure 11, this description is not clear enough.
Author Response
Response Letter
Manuscript ID: micromachines-2296809
Manuscript Title: Multiband Rectenna Using ZnO Based Planar Schottky Diode for RF Energy Harvesting Applications
Type of manuscript: Article
Authors: Somaya Kayed, Dalia M. Elsheakh*, Hesham A Abdel Hady, Heba A. Shawkey
Dear Editor and Reviewers:
First, we would like to sincerely thank the editor and the reviewers for their valuable comments. We have prepared detailed responses for reviewer’ comments. In the revised paper, we use highlighted yellow text for modified sections in the paper based on the reviewer’ comments.
Sincerely yours,
Comments of Editor Comments:
This work reported a Multiband Microstrip Rectenna Using ZnO Based Planar Schottky Diode for RF Energy Harvesting Applications. An UWB antenna operating in 2.5 – 25 GHz range with a multi-band rectifier design is studied.
Comment 1: The abstract can be more concise, and the content does not need to be written using the software name, and can be placed in the main text. Units are not superscripted correctly.
Response: The abstract is become more concise and removed any redundant sentence.
Comment 2: The author designed the load to be 300 ohms, can it have better performance on other loads?
Response: Different load resistances are simulated (300 Ω, 1K Ω and 5K Ω). The 300 Ω load has the largest DC output voltage, so it is selected for fabrication.
This paragraph is added at the end of section 3.2 (the highlighted paragraph to the paper) as follow:
“From previous simulation, the 300Ω load is selected for rectifier fabrication as it has the largest DC output. The rectifier has a considered DC output voltage that can reach 750 mV, which is sufficient for biasing tiny sensors and portable electronics devices.”
Comment 3: Figure 11 can provide a system architecture diagram, and restate enough details and definitions in Figure 11, this description is not clear enough.
Response: Measurements’ setup block diagram is added in the Fig. 11(a) as shown below:
(a)
(b)
- (d)
Figure 11. The setup of the RF energy harvesting (a) Configuration , (b) Measured of proposed system, (c) Output volt and power at 300 ohm and (d) Comparison of simulated and measured efficiency .

Round 2
Reviewer 1 Report (New Reviewer)
The author response is noted.
However, few more observations are as follows:
1. Antenna 3D patterns are missing.
2. Fig. 9(c) should also have result without Ag/ZnO material. The multi-band operation of this rectifier also needs to be addressed. A statement stating simply due to diode will not work.
3. A comparison of different rectifier topologies based on planar Ag/ZnO Schottky diode is needed with the proposed work.
4. It is also essential to prove the merit of proposed rectifier design with other works. The table 6 present that the literature works are achieving better results than the proposed work.
5. Check grammatical and typo mistakes throughout the paper.
Author Response
Response Letter
Manuscript ID: micromachines-2296809
Manuscript Title: Multiband Rectenna Using ZnO Based Planar Schottky Diode for RF Energy Harvesting Applications
Type of manuscript: Article
Authors: Somaya Kayed, Dalia M. Elsheakh*, Hesham A Abdel Hady, Heba A. Shawkey
Dear Editor and Reviewers:
First, we would like to sincerely thank the editor and the reviewers for their valuable comments. We have prepared detailed responses for reviewer’ comments. In the revised paper, we use highlighted yellow text for modified sections in the paper based on the reviewer’ comments.
Sincerely yours,
Comments of Editor Comments:
Comment 1: Antenna 3D patterns are missing.
Response: Thank you very much for your efforts, we added the simulated 3D radiation pattern in manuscript text and table 3 as follow:
“The simulated three-dimension radiation pattern of the proposed monopole antenna at theses frequency is shown in table 3. The comparison between measured and simulated radiation pattern obtained of the proposed antenna at 3.5 GHz,12.5 GHz, 14.5 GHz, and 17.5 GHz in the X-Z plane and X-Y plane of the Phi = 0o (H-plane) and phi = 90o (E-plane), respectively are shown in Table 3.”
Table 3. Far field characteristics of proposed UWB antenna at different frequencies in the XZ and XY planes (Simulated (solid line) – Measured (dash line) with the simulated 3D radiation pattern.
|
F(GHz) |
Phi=0 |
3D Radiation |
Phi=90 |
3D Radiation |
|
3.5 |
|
|
|
|
|
12.5 |
|
|
|
|
|
14.5
|
|
|
|
|
|
17.5
|
|
|
|
|
Comment 2: Fig. 9(c) should also have result without Ag/ZnO material. The multi-band operation of this rectifier also needs to be addressed. A statement stating simply due to diode will not work.
Response :Figure 9(c) is modified and the response of the rectifier circuit without the proposed diode is simulated as shown in the figure.
- (b)
(c)
Figure 9. Shunt half-wave rectifier (a) Circuit diagram, (b) Photo of fabricated, (c) S-parameter.
This figure is mention in the manuscript as follow:
“Figure 9(c) shows the simulated rectifier without diode and the comparison between simulation and measurements of complete rectifier circuit, where the difference could be explained due to the effect of the DC pass filter load on the rectifier, while the S-parameter of the diode (used for rectifier design) is measured with just effect of T.L3 and T.L is considered.”
Comment 3: A comparison of different rectifier topologies based on planar Ag/ZnO Schottky diode is needed with the proposed work.
Response: Thank you for your comment; however, most of publications in this field are interested in the study of diode performance alone. Therefore, the authors could not find different rectifier topologies with Zno schottky diode.
Comment 4: It is also essential to prove the merit of proposed rectifier design with other works. The table 6 present that the literature works are achieving better results than the proposed work.
Response: The most obvious merit for the proposed devise over the other devices is simple implementation technique. Moreover, no need for sophisticated fabrication steps or expensive fabrication equipment and allows implementation for a complete system with no need for off-shelf components.
Comment 5: Check grammatical and typo mistakes throughout the paper.
Response: The authors have revised the manuscript according to the comments.
Reviewer 2 Report (New Reviewer)
The authors have revised the manuscript according to the comments.
1. One of the key contribution of the manuscript is the self-made diode. Please add some details or discussions on the diode, e.g., whether the Schottky barrier is between the ZnO and Cu or between the ZnO and Ag, whether doping was applied to ZnO, how much is the breakdown voltage of the diode, and so on.
2. How to control the size of the diode? Say, how to make the Schottky barrier to be 1mm by 1mm.
Author Response
Response Letter
Manuscript ID: micromachines-2296809
Manuscript Title: Multiband Rectenna Using ZnO Based Planar Schottky Diode for RF Energy Harvesting Applications
Type of manuscript: Article
Authors: Somaya Kayed, Dalia M. Elsheakh*, Hesham A Abdel Hady, Heba A. Shawkey
Dear Editor and Reviewers:
First, we would like to sincerely thank the editor and the reviewers for their valuable comments. We have prepared detailed responses for reviewer’ comments. In the revised paper, we use highlighted yellow text for modified sections in the paper based on the reviewer’ comments.
Sincerely yours,
Comments of Editor Comments:
Comment 1: One of the key contribution of the manuscript is the self-made diode. Please add some details or discussions on the diode, e.g., whether the Schottky barrier is between the ZnO and Cu or between the ZnO and Ag, whether doping was applied to ZnO, how much is the breakdown voltage of the diode, and so on.
Response: The next paragraph is added in the update manuscript “Section 3”
“Among semiconductor devices used to implement thin film schottky diodes, ZnO is a promising candidate for use in printed diodes, providing a potentially cost-effective and scalable solution for a range of electronic applications. Besides, its high breakdown voltage (>15V), high rectification ratio compared with other diodes, make this environment-friendly process a serious option for power electronics and energy-harvesting [24]. In this paper ZnO n-type semiconductor with Ag electrode is used to implement PSD for multiband operation with total dimensions 1×1 mm2. The proposed diode consists of Ag/ZnO layers supported on a copper layer with the schottky barrier at Ag/ZnO junction. In this paper ZnO n-type semiconductor with Ag electrode is used to implement PSD for multiband operation with total dimensions 1×1 mm2. “
Comment 2: How to control the size of the diode? Say, how to make the Schottky barrier to be 1mm by 1mm.
Response: Size of the diode is controlled as follows:
- Fixing a small piece of an adhesive tape on the copper transmission line where the PSD is implemented.
- According to the required area, part of this tape is removed by laser cutting ( using laser source available in Lab.)
- After deposition of different layers of the diode the remaining part of the tape is removed.

Reviewer 3 Report (New Reviewer)
Currently no comments, suggestions can be published.
Author Response
Thank you very much!
Round 3
Reviewer 1 Report (New Reviewer)
Author comments are noted.
Author Response
Thank you for your review
This manuscript is a resubmission of an earlier submission. The following is a list of the peer review reports and author responses from that submission.
Round 1
Reviewer 1 Report
The Authors present a “Multiband Rectenna Using ZnO Based Planar Schottky Diode for RF Energy Harvesting Applications”. The antenna is composed of a monopole antenna and a rectifier circuit. For all the above, publication of this manuscript is not recommended.
- The proposed design of the monopole antenna is not new. The performance of the antenna in terms of adaptation to -10 dB can be considerably improved.
- Reading the document, we note the absence of a design equation. It would be nice if the author could provide some equations for determining the different dimensions.
- The curve of the measured reflection coefficient in Figure 5(c) shows that the antenna is mismatched at low frequencies between 5GHz and 9GHz (at -10 dB) and at high frequencies above 20 GHz. The difference between simulation and measurement results is quite large. This mismatch implies that the antenna does not receive well in these working bands. On the other hand, we cannot say that the antenna is UWB (3.1GHz-11.6Ghz) since generally, it requires an adaptation at -10dB over the entire band mentioned.
- For the radiation pattern, we note that the shape of the radiation for the low frequencies of the working band (Table 3) is not symmetrical in the plane phi = 90°. This can be explained by the shape of the radiation patch which is not symmetrical with respect to the Y-axis. This architecture makes the antenna a little more directive on one side compared to the other side which is the opposite as shown in the radiation pattern figure at 3.5 GHz. We also notice that the omnidirectional radiation that usually characterizes monopole antennas is absent in this case.
- How do you explain the significant difference between the Gain curves in Figure 7(a)
- In table 4, the author notes that the working band is from 2.5GHz up to 25 GHz (single band), and at the same time he presents it as a dual-band antenna. There is a contradiction in the statements.
- The references used in Table 4 are quite old (2017, 2014). The author should take as a model of comparison more recent works (2020, 2021, 2022).
- In figure 8, there is an error in the wording of the title, (b) and (c) are reversed.
- In figure 9, the comparison of the results shows a fairly large difference from 6 GHz. How do you explain this result?
- In figure 11, the simulated efficiency is very low, how do you explain this result? What are the consequences on the proposed design?
- Table 6 must be redone, it is difficult to discern the information of each reference that is mixed. The table should be more spaced out. Regarding the references, the same remark, they are quite old. The author should choose more recent work to compare with his work.
- English grammar and readability can be improved
- The presentation of this manuscript can be improved. There are a lot of punctuation errors and typos
Author Response
Response Letter
Manuscript ID: micromachines-2072992
Manuscript Title: Multiband Rectenna Using ZnO Based Planar Schottky Diode for RF Energy Harvesting Applications
Type of manuscript: Article
Authors: Somaya Kayed, Dalia M. Elsheakh*, Hesham A Abdel Hady, Heba A.
Shawkey
Dear Editor and Reviewers:
First, we would like to sincerely thank the editor and the reviewers for their valuable comments. We have considered all the reviewers’ comments and suggestions and have modified our manuscript accordingly.
We have also prepared detailed responses for reviewers’ comments. In the revised paper, we use highlighted yellow text for modified sections, figures & tables in the paper based on the reviewers’ comments.
Sincerely yours,
Comments of Reviewer 1
The Authors present a “Multiband Rectenna Using ZnO Based Planar Schottky Diode for RF Energy Harvesting Applications”. The antenna is composed of a monopole antenna and a rectifier circuit. For all the above, publication of this manuscript is not recommended.
Response: We deeply appreciate to the reviewer for his/her positive evaluation and we hope that the new update manuscript gains your appraisal.
Comments 1: The proposed design of the monopole antenna is not new. The performance of the antenna in terms of adaptation to -10 dB can be considerably improved.
Response: Indeed, this is correct, using the monopole antenna is not a new idea; however, what is new is the shape of the proposed antenna with different technology used to enhance the operating bandwidth. The proposed monopole antenna has impedance bandwidth at -6 dB according to IEEE standard (due to the huge revolution in the electronic circuit sensitivity) extend from 2.5 GHz up to 24 GHz. While, if we used -10 dB (VSWR ≤ 2) the characteristic impedance bandwidth of the proposed monopole antenna start to resonant at 3 GHz to 6 GHz and from 10 GHz to 18 GHz. Ref. [29] is added that show the VSWR≤2.5 is accepted.
(a) (b)
Figure 5, Manufactured prototype of the developed UWB antenna (a) front part, (b) back part, the Simulated and measured of the proposed antenna of the UWB antenna structure (c) |S11| and (d) VSWR.
Comments 2: Reading the document, we note the absence of a design equation. It would be nice if the author could provide some equations for determining the different dimensions.
Response : Thank you for this comment and we added a paragraph that explains the initial design of the monopole antenna in section 2.2
The resonance frequency of clipart moon shaped radiating patch could be calculated using Eq. (1) and (2) according to the standard formula [8], [9], the resonance frequency (fr) is expressed as:
(1)
Where () is the velocity of light, () is resonant of frequency, D1 is the outer diminution of clipart moon shaped () is dielectric constant of substrate.
The effective dielectric constant () could be expressed as:
(2)
Where (h) is the height of the substrate.
Comments 3: The curve of the measured reflection coefficient in Figure 5(c) shows that the antenna is mismatched at low frequencies between 5GHz and 9GHz (at -10 dB) and at high frequencies above 20 GHz. The difference between simulation and measurement results is quite large. This mismatch implies that the antenna does not receive well in these working bands. On the other hand, we cannot say that the antenna is UWB (3.1GHz- 11.6Ghz) since generally, it requires an adaptation at -10 dB over the entire band mentioned.
Response: The differences between simulated and measured results at low and high frequencies could be attributed for some reasons. Firstly, the effects of the coaxial cable used in the measurement. Secondly, SMA solder used in soldering the SMA conductor to the feeder line. Lastly, other electromagnetic interference signals in the atmosphere and ideal model uses for simulation as well as manufacturing and measurement tolerances in the positioned antenna I considered as one of the most important reasons. IEEE standard changes the regulation of the confirmed characteristic impedance bandwidth from -10 dB to -6 dB. Due to the amazing development in electronic circuits, the sensitivity and accuracy of signal extraction have increased. In addition, there are many published papers using this threshold.
Comments 4: For the radiation pattern, we note that the shape of the radiation for the low frequencies of the working band (Table 3) is not symmetrical in the plane phi = 90°. This can be explained by the shape of the radiation patch, which is not symmetrical with respect to the Y-axis. This architecture makes the antenna a little more directive on one side compared to the other side which is the opposite as shown in the radiation pattern figure at 3.5 GHz. We also notice that the omnidirectional radiation that usually characterizes monopole antennas is absent in this case.
Response: Yes, you are completely right about the radiation pattern, even though the far field radiation pattern is different with the resonant frequency compared to the main element of antenna contributed in the radiation. According to the current over the surface distribution of the monopole, the radiation pattern is changed. We added a paragraph in section 2.5 that explains this concept.
“ As well as it is noted that the shape of the radiation for the low frequencies at 3.5 GHz is not symmetrical in the plane phi = 90° due to the shape of the radiation patch, which is not symmetrical with respect to the Y-axis. This configuration makes the antenna a little more directive on one plane compared to the other plane, which is the opposite as shown in the radiation pattern. In addition, at high frequency the radiation pattern becomes Omni radiation pattern, because of the far field radiation pattern that is different with the resonant frequency according to the main element of antenna contributed in the radiation. According to the current over the surface distribution of the monopole, the radiation pattern is changed.”
Comments 5: How do you explain the significant difference between the Gain curves in Figure 7(a)
Response: It is indeed true that there is a difference between two simulators used to design the proposed monopole antenna as HFSS and CST electromagnetic simulators. There are several reasons behind that. First and foremost, the type of numerical method used to analyse the structure. HFSS simulator used finite element method (FEM), while CST used the finite integration technique (FIT) and the transmission line matrix (TLM). Furthermore, the dimension of the radiation pattern used in the design of the proposed monopole antenna in HFSS is fixed and not less than by l/4 of resonant frequency, while in CST, it is automatically assigned from the simulator. We mentioned this in the section 2.5, Far-Field Properties, page 8.
“There is difference between the two electromagnetic simulators used to design the proposed monopole antenna as HFSS and CST. the type of numerical method used to analyze the structure is one of the major reasons. HFSS used finite element method (FEM), while CST used the finite integration technique (FIT) and the transmission line matrix (TLM). Secondly, the dimension of the radiation pattern used to design the proposed monopole antenna in HFSS is fixed and not less than by l/4 of operating resonant frequency at single frequency, while in CST, it is automatically, assign from simulator and adaptive with frequency. ”
Comments 6: In table 4, the author notes that the working band is from 2.5 GHz up to 25 GHz (single band), and at the same time he presents it as a dual-band antenna. There is a contradiction in the statements.
Response: We apologize for this typo, and the word has been standardized throughout the article for “UWB”
Comments 7: The references used in Table 4 are quite old (2017, 2014). The author should take as a model of comparison more recent works (2020, 2021, 2022).
Response: Thank you for this valuable comment and Table 4 has been updated
|
Ref. |
Size (mm3) |
Size (lg)2 |
Substrate
|
Res. Fre./BW (GHz) |
No. of bands |
|S11| (dB) |
Application |
|
[14] |
35 |
0.34×0.36 |
RT = 2.2 |
2.22–22 |
1 |
-18 |
UWB |
|
[15] |
501.57 |
0.42×0.54 |
FR4 =2.2 |
7.5-9 |
1 |
-19 |
UWB |
|
[16] |
40×38×0.8 |
1.26×0.86 |
RT = 2.2 |
2.85-12 |
1 |
-18 |
UWB |
|
[17] |
1.6 |
1.11×0.36 |
RT =2.2 |
2.6, 11.2 |
2 |
-21 |
C-band Satellite Communication |
|
[18] |
01.610 |
0.66×0.66 |
FR4 =4.4 |
2.2-12 |
1 |
-21 |
UWB |
|
This Work |
32 ×31×1.575 |
0.31×0.21 |
RT4003 = 2.2 |
2.5 up to 25 |
2 |
-25 |
5G & IoT Communication |
14- S. Ullah, C. Ruan, M. S. Sadiq, T. U. Haq, andW. He, ``High ef_cient and ultra-wide band monopole antenna for microwave imaging and communication applications,'' Sensors, vol. 20, no. 1, p. 115, Dec. 2019.
15- Y. Yuan, X. Xi, and Y. Zhao, ‘‘Compact UWB FSS reflector for antenna gain enhancement,’’ IET Microw., Antennas Propag., vol. 13, no. 10, pp. 1749–1755, Aug. 2019.
16 S. Luo, Y. Chen, D Wang, and L. Yejun " A monopole UWB antenna with sextuple band-notched based on SRRs and U-shaped parasitic strips", AEU, International Journal of Electronics and Communications, Vol.99, pp. 59 – 69, 2020.
17- S. Mukherjee, A. Roy and S. Bhunia, “Design of compact UWB slotted hexagonal monopole antenna with 3.5/5.5 GHz dual band rejection,” Journal of Nano-and Electronic Physics, vol. 13, no. 3, pp. 03026-1–03026-4, 2021.
18- A. J.A.Al-Gburi, I. B. M. Ibrahim, Z. Zakaria, B. H. Ahmad, N. A. Bin Shairi et al., “High gain of UWB planar antenna utilising FSS reflector for UWB applications,” Computers, Materials & Continua, vol. 70, no. 1, pp. 1419–1436, 2022.
Comments 8: In figure 8, there is an error in the wording of the title; (b) and (c) are reversed.
Response : Thank you, the error is corrected.
- (c)
Figure 8, Ag/ZnO PSD (a) structure, (b) Measured I/V characteristics and (c) Measured S-parameter.
Comments 9: In figure 9, the comparison of the results shows a fairly large difference from 6 GHz. How do you explain this result?
Response: This can be explained due to the effect of the DC pass filter load on the rectifier, while the S-parameter of the diode is measured with just effect of T.L.3 and T.L. are considered.”
This paragraph is added to section 3.2
Comments 10: In figure 11, the simulated efficiency is very low, how do you explain this result? What are the consequences on the proposed design?
Response: The efficiency shown in figure 11 is calculated from measured DC voltage. Its low value can be explained due to the usage of commercial materials for diode implementation, as Ag electrode and copper tape used to connect the Ag electrode with the copper transmission lines- by using higher grade materials can increase the efficiency. Besides, the Ag electrode is dropped manually on the ZnO surface, applying advanced deposition techniques as thermal evaporation (which is not available for authors) can improve diode performance. and Ref. [28] mentioned flexible substrate, but they didn’t mention the antenna design.
This paragraph is added to section 3.3
“Also a single matching circuit is designed for multiband operation, this reduces the efficiency compared with single frequency matching circuit but has a good impact on rectifier area.”
What are the consequences on the proposed design?
The overall performance is low compared with other similar designs, but the proposed design is a proof of concept for rectenna implementation with simple Lab deposition techniques.
Comments 11: Table 6 must be redone; it is difficult to discern the information of each reference that is mixed. The table should be more spaced out. Regarding the references, the same remark, they are quite old. The author should choose more recent work to compare with his work.
Response : It is done
Regarding the references, the same remark, they are quite old. The author should choose more recent work to compare with his work.
Most references show implementation and characterization of a thin film Schottky diode only or fabrication of a rectenna with commercial diodes. Few literatures show rectenna implementation with in-situ Schottky diodes.
Reference [28] is added but antenna design is not presented.
Comments 12: English grammar and readability can be improved and the presentation of this manuscript can be improved. There are a lot of punctuation errors and typos
Response: According to reviewers’ comments, we have revised the manuscript. We hope that the main theme and contribution of this work is more reachable in the present form to the reviewer for his/her positive evaluation.

Reviewer 2 Report
The design and implementation of a multiband UWB antenna for an RF energy harvesting application were discussed in this research. The manuscript has the potential to be accepted for publication. However, several updates are required before resubmission.
1. From Figure 11 (c) it can be observed that the overall efficiency of the proposed system is very low. Explain more about the proposed system’s efficiency value, and compare the proposed efficiency with other state-of-the-art works in the comparison table to show how the proposed system is better than other systems reported in the literature.
2. Add more recent similar works in the comparison table to show the design significance of this proposed system.
3. What kind of polarization is generated by the antenna? Please mention this in the manuscript.
4. The caption of Figure 8 is incorrect. Please check and update accordingly.
5. Explain more about the benefit of the multiband UWB antenna in the introduction section. Here are some suggestions. Isolation and Gain Improvement of a Rectangular Notch UWB-MIMO Antenna, Sensors, 2022. A Novel Meander Bowtie-Shaped Antenna with Multi-Resonant and Rejection Bands for Modern 5G Communications, Electronics, 2022. The authors also can find related articles by internet search.
6. Try to avoid special characters (e.g. @) in the abstract.
7. Figure 1 and Figure 9 (a) are not clear, please provide a better-quality sketch.
8. It is recommended to provide a 2D antenna geometry and configuration in the manuscript in Figure 2 for a better representation. And the substrate height h is not shown in Figure 2.
9. Choose different line types to represent data in Figure 5 and Figure 7 especially for Figure 5(d) and Figure 7 (a), those data representation is very poor.
10. In Table 6, “Flexible substrate” is not appropriate for reference 20 at the Antenna Structure column. Please check and update accordingly.
Author Response
Response Letter
Manuscript ID: micromachines-2072992
Manuscript Title: Multiband Rectenna Using ZnO Based Planar Schottky Diode for RF Energy Harvesting Applications
Type of manuscript: Article
Authors: Somaya Kayed, Dalia M. Elsheakh*, Hesham A Abdel Hady, Heba A.
Shawkey
Dear Editor and Reviewers:
First, we would like to sincerely thank the editor and the reviewers for their valuable comments. We have considered all the reviewers’ comments and suggestions and have modified our manuscript accordingly.
We have also prepared detailed responses for reviewers’ comments. In the revised paper, we use highlighted yellow text for modified sections, figures & tables in the paper based on the reviewers’ comments.
Sincerely yours,
Comments of Reviewer 2
The design and implementation of a multiband UWB antenna for an RF energy harvesting application were discussed in this research. The manuscript has the potential to be accepted for publication. However, several updates are required before resubmission.
Response: Thank you for your constructive comments and suggestions. We have added all reviewers’ consideration into in the revised manuscript.
Comments 1: From Figure 11 (c) it can be observed that the overall efficiency of the proposed system is very low. Explain more about the proposed system’s efficiency value, and compare the proposed efficiency with other state-of-the-art works in the comparison table to show how the proposed system is better than other systems reported in the literature
Response: “Its low value can be explained due to the usage of commercial materials for diode implementation, as Ag electrode and copper tape used to connect the Ag electrode with the copper transmission lines- by using higher grade materials can increase the efficiency. Besides, the Ag electrode is dropped manually on the ZnO surface, applying advanced deposition techniques as thermal evaporation (which is not available for authors) can improve diode performance.”
This paragraph is added to section 3.3
Comments 2: Add more recent similar works in the comparison table to show the design significance of this proposed system.
Response: Most references show implementation and characterization of a thin film Schottky diode only or fabrication of a rectenna with commercial diodes. Few literatures show rectenna implementation with in-situ Schottky diodes. Reference [28] is added to table 6.
Comments 3: What kind of polarization is generated by the antenna? Please mention this in the manuscript.
Response: The proposed monopole antenna is linearly polarized and it is mention in different manuscript sections as abstract, introduction and Antenna design section.
Comments 4: The caption of Figure 8 is incorrect. Please check and update accordingly.
Response : Thank you, the error is corrected (b) and (c) are exchanged.
- (c)
Figure 8, Ag/ZnO PSD (a) structure, (b) Measured I/V characteristics and (c) Measured S-parameter.
Comments 5: Explain more about the benefit of the multiband UWB antenna in the introduction section. Here are some suggestions. Isolation and Gain Improvement of a Rectangular Notch UWB-MIMO Antenna, Sensors, 2022. A Novel Meander Bowtie-Shaped Antenna with Multi-Resonant and Rejection Bands for Modern 5G Communications, Electronics, 2022. The authors also can find related articles by internet search.
Response: Thank you for your comments. We demonstrated it in introduction section. Also in [6] the MIMO configuration of the proposed antenna shows and ultra-wideband (UWB) 5G communications is best suited for an end-user customer to accept and reject different communication services by just utilizing the corresponding band. The UWB- MIMO isolation and gain of the antenna are enhanced by using a parasitic isolator and the surface is also shown radiator in the form of an arc with a lower ground plane composed of parasitic components in [7, 8] respectively.
Added references:
6- Y. S. Faouri , S. Ahmad, P. Naser, H. Chan and R. Abd-Alhameed “A Novel Meander Bowtie-Shaped Antenna with Multi-Resonant and Rejection Bands for Modern 5G Communications” Electronics 2022, 11, 821. [CrossRef]
7- A. Abbas, N. Hussain, M. AbuSufian, J. Jung, S. Park and N. Kim * “Isolation and Gain Improvement of a Rectangular Notch UWB-MIMO Antenna” Sensors 2022, 22, 1460. [CrossRef]
Comments 6: Try to avoid special characters (e.g. @) in the abstract.
Response: We deeply appreciate the reviewer for his/her positive evaluation and valuable comments. According to reviewers’ comments, we have revised the manuscript.
Comments 7: Figure 1 and Figure 9 (a) are not clear, please provide a better quality sketch.
Response: Thank you. It has drawn again with improved quality.
Figure 9(a) Figure 9(b)
Comments 8: It is recommended to provide a 2D antenna geometry and configuration in the manuscript in Figure 2 for a better representation and the substrate height h is not shown in Figure 2.
Response : We provide a 2D Geometry of the proposed antenna with 3D as in Figure 2.
- (b)
Figure 2, Proposed monopole Antenna (a) 2D Geometry and (b) 3D Geometry.
Comments 9: Choose different line types to represent data in Figure 5 and Figure 7 especially for Figure 5(d) and Figure 7 (a), those data representation is very poor.
Response: Thank you, these figures have been corrected.
(c) (d)
Figure 5, Manufactured prototype of the developed UWB antenna (a) front part, (b) back part, the Simulated and measured of the proposed antenna of the UWB antenna structure (c) |S11| and (d) VSWR.
- (b)
Figure 7, Simulated and measured (a) peak gain and (b) radiation efficiency versus frequency.
Comments 10: In Table 6, “Flexible substrate” is not appropriate for reference 20 at the Antenna Structure column. Please check and update accordingly.
Response: The antenna structure is not set in the paper. It is -mentioned as flexible Wi-Fi band antenna.

Reviewer 3 Report
1. The authors need to mention the novelties of this work in the last section of the manuscript according to point wise discussion
2. A few more recent works on rectenna are required to be included.
3. The authors need to involve the necessary design for the antenna
4. In Figure 3, the role of antenna 3 is not clear in the design evolution as compared to antenna 2. why authors scientifically adopted antenna 3?
5. Table 2, current distributions should not be shown in form of table, it must be presented in form of figure with magnitude scaling
6. Concerning Figure 5, there are changes in measured data as validated from S11 and VSWR , specially from 20 to 24 GHz. The authors must verify VSWR data.
7. More explanation on radiation patterns are required to be given.
8. As per figure 7 , is it possible that measured gain will be higher than simulated one, CST results up to 11 GHz. please clarify and explain.....
Author Response
Response Letter
Manuscript ID: micromachines-2072992
Manuscript Title: Multiband Rectenna Using ZnO Based Planar Schottky Diode for RF Energy Harvesting Applications
Type of manuscript: Article
Authors: Somaya Kayed, Dalia M. Elsheakh*, Hesham A Abdel Hady, Heba A.
Shawkey
Dear Editor and Reviewers:
First, we would like to sincerely thank the editor and the reviewers for their valuable comments. We have considered all the reviewers’ comments and suggestions and have modified our manuscript accordingly.
We have also prepared detailed responses for reviewers’ comments. In the revised paper, we use highlighted yellow text for modified sections, figures & tables in the paper based on the reviewers’ comments.
Sincerely yours,
Comments of Reviewer 3
The authors need to mention the novelties of this work in the last section of the manuscript according to point wise discussion
Response: Thank you for your valuable comments.
Comments 1: A few more recent works on rectenna are required to be included
Response: Most references show implementation and characterization of a thin film schottky diode only or fabrication of a rectenna with commercial diodes. Few literatures show rectenna implementation with in-situ Schottky diodes. Reference [28] is added to table 6 and mention that but they didn’t mention the antenna design.
Comments 2: The authors need to involve the necessary design for the antenna
Response: Thank you for this comment. We added a paragraph that explains the initial design of the monopole antenna in section 2.2
The resonance frequency of clipart moon shaped radiating patch could be calculated using Eq. (1) and (2) according to the standard formula [8], [9], the resonance frequency (fr) is expressed as:
(1)
Where ( ) is the velocity of light, ( ) is resonant of frequency, D1 is the outer diminution of clipart moon shaped ( ) is dielectric constant of substrate.
The effective dielectric constant ( ) could be expressed as:
(2)
Where (h) is the height of the substrate.
Comments 3: In Figure 3, the role of antenna 3 is not clear in the design evolution as compared to antenna 2. why authors scientifically adopted antenna 3?
Response: antenna 3 in the design steps is etched rectangular slot in the ground plane under the 50 ohm transmission line. Also, we designed it to improve the antenna matching.
Comments 4: Table 2, current distributions should not be shown in form of table, it must be presented in form of figure with magnitude scaling
Response: About table 2, it is very helpful to clarify the two man plane E and H plane at different resonant frequency and about the scale (all figure in E plane and H plane haves same scale).
Comments 5: Concerning Figure 5, there are changes in measured data as validated from S11 and VSWR, specially from 20 to 24 GHz. The authors must verify VSWR data.
Response: Thank you very much for your comment. In the ERI Lab, we saved two separate excel file one for |S11| and other for VSWR. We didn’t extract the VSAR from S11 from Eq.
VSWR= 1-|S11|/1+|S11|
Comments 6: More explanation on radiation patterns are required to be given.
Response: We added three new paragraphs about radiation pattern in section 2.5 as follow:
“As well as it is noted that the shape of the radiation for the low frequencies at 3.5 GHz is not symmetrical in the plane phi = 90° due to the shape of the radiation patch, which is not symmetrical with respect to the Y-axis. This configuration makes the antenna a little more directive on one plane compared to the other plane, which is the opposite as shown in the radiation pattern. In addition, at high frequency the radiation becomes an omniradiation pattern, because of the difference of the far field radiation pattern comparing to the resonant frequency according to the main element of antenna contributed in the radiation. According to the current over the surface distribution of the monopole, the radiation pattern is changed.”
Comments 7: As per figure 7 , is it possible that measured gain will be higher than simulated one, CST results up to 11 GHz. please clarify and explain.....
Response: The measured peak gain of proposed antenna is larger than the simulated one in some frequency points. This is because the directivity is changed during processing and measuring, can the measured result be larger than simulated one due to the power concentration. A third problem arises from the test equipment itself: it is impossible to avoid the interactions introduced by the test equipment and test fixtures and/or equipment holders. This is because it needs to be performed in an open space environment withwell controlled minimum reflections. There are three categories of widely used antenna pattern and gain measurement systems: Far Field Range, Near Field Range, and Compact Range. Generally these systems are unwieldly and expensive, regardless of the frequency band. If a Vector Network Analyzer (VNA) or Synthesizer and Spectrum Analyzer are added, the cost becomes unaffordable. As well as there is difference between two simulators used to design the proposed monopole antenna as HFSS and CST electromagnetic simulators. This because many reasons, first due to the type of numerical method .used to analyse the structure. HFSS used finite element method (FEM) while CST used the finite integration technique (FIT) and the transmission line matrix (TLM). Second, the dimension of the radiation pattern used in the design the proposed monopole antenna in HFSS is fixed and not less than by l/4 of resonant frequency. While in CST it is automatically, assign from simulator.

Round 2
Reviewer 1 Report
Following the various changes made to the manuscript by the authors. I am in favor of a publication in the micromachines journal.
Author Response
Response Letter
Manuscript ID: micromachines-2072992
Manuscript Title: Multiband Rectenna Using ZnO Based Planar Schottky Diode for RF Energy Harvesting Applications
Type of manuscript: Article
Authors: Somaya Kayed, Dalia M. Elsheakh*, Hesham A Abdel Hady, Heba A. Shawkey
Dear Editor and Reviewers:
First, we would like to sincerely thank the editor and the reviewers for their valuable comments. We have considered all the reviewers’ comments and suggestions and have modified our manuscript accordingly.
We have also prepared detailed responses for reviewers’ comments. In the revised paper, we use highlighted yellow text for modified sections, figures & tables in the paper based on the reviewers’ comments.
Sincerely yours,
Comments of Reviewer 1
Comments : Following the various changes made to the manuscript by the authors. I am in favor of a publication in the micromachines journal.
Response : Thank you very much.

Reviewer 2 Report
All comments have been addressed in this version. It has merit to be published in Micromachines by checking grammatical errors and typos.
Author Response
Response Letter
Manuscript ID: micromachines-2072992
Manuscript Title: Multiband Rectenna Using ZnO Based Planar Schottky Diode for RF Energy Harvesting Applications
Type of manuscript: Article
Authors: Somaya Kayed, Dalia M. Elsheakh*, Hesham A Abdel Hady, Heba A.
Shawkey
Dear Editor and Reviewers:
First, we would like to sincerely thank the editor and the reviewers for their valuable comments. We have considered all the reviewers’ comments and suggestions and have modified our manuscript accordingly.
We have also prepared detailed responses for reviewers’ comments. In the revised paper, we use highlighted yellow text for modified sections, figures & tables in the paper based on the reviewers’ comments.
Sincerely yours,
Comments of Reviewer 2
Comments : All comments have been addressed in this version. It has merit to be published in Micromachines by checking grammatical errors and typos.
Response : thank you very much.

Reviewer 3 Report
There are many literature available on rectenna but still the authors didnt included after the advice given in 1st review process.
The authors need to involve the necessary design equations for the antenna-- For this comment, the authors need to show the detailed calculation from the provided equations and they must comment how the design parameters dimensions are finalized?
In Figure 3, the role of antenna 3 is not clear in the design evolution as compared to antenna 2. why authors scientifically adopted antenna 3?
The answers authors provided for this question are not at all satisfactory.
The answers given by authors for comments 5, and 7 are not at all logical. The authors need to show VNA measurement set up image.
The figures must be straight line for better clarity.
The radiation pattern data seems quite surprising, So much close agreement with simulation is not possible?????
Author Response
Response Letter
Manuscript ID: micromachines-2072992
Manuscript Title: Multiband Rectenna Using ZnO Based Planar Schottky Diode for RF Energy Harvesting Applications
Type of manuscript: Article
Authors: Somaya Kayed, Dalia M. Elsheakh*, Hesham A Abdel Hady, Heba A. Shawkey
Dear Editor and Reviewers:
First, we would like to sincerely thank the editor and the reviewers for their valuable comments. We have considered all the reviewers’ comments and suggestions and have modified our manuscript accordingly.
We have also prepared detailed responses for reviewers’ comments. In the revised paper, we use highlighted yellow text for modified sections, figures & tables in the paper based on the reviewers’ comments.
Sincerely yours,
Comments of Reviewer 3
Comments 1: There are many literature available on rectenna but still the authors didnt included after the advice given in 1st review process.
Response: The authors included 4 references to table 6 (References [29-32]) but with commercial Schottky diodes are used for rectenna implementation.
|
Ref |
Diode structure |
Antenna structure |
Max. effic. |
DC output Volt
|
Frequency (GHz) |
|
[20] |
MoS2 Schottky barrier |
Flexible substrate |
40.1% |
0.9 V/10GHz, 0.5V/15GHz (pin=0dBm) |
2.4, 5.9, 10 & 15 |
|
[25] |
Ni-NiO-Cr metal–insulator–metal (MIM) |
microstrip slot |
N/A |
56mV (pin=0dBm) |
2.45 |
|
[26] |
Au/HfO2/Pt (MIM) |
Bow Tie |
N/A |
0.25mV (at pin -20dBm) |
55-65 |
|
[27] |
Graphene |
Patch array |
N/A |
95mV |
28 |
|
[28] |
MoS2 Schottky barrier |
Commercial antenna |
N/A |
10mV/2.4GHz (pin=0dBm) |
0.1-10 |
|
[29] |
SMS7630 |
Meander monopole |
40% |
N/A |
2.4 |
|
[30] |
SMS7630& MA4E-1319 |
Broadband antipodal Vivaldi |
12% |
6.5V (pin=20 dBm) |
22.6-40 |
|
[31] |
SMS7630 |
Ddual-ring shaped monopole |
40% |
41mV (pin=0dBm)
|
1.8-2.6 |
|
[32] |
HSMS 2860 |
Dual linearly polarized antenna array |
40% |
500mV |
1.8-2.1 |
|
This work |
ZnO Schottky barrier |
Moon shaped cut |
15% |
250mV/3.5GHz (pin=0dBm) |
3.5,6,8,10,18 |
- A. Eid, A.; Hester, J.; Nauroze. A.; Lin, T.; Costantine, J.; Tawk, Y.; Ramadan, A.; and Tentzeris, M.; “A Flexible Compact Rectenna for 2.4GHz ISM Energy Harvesting Applications,” IEEE International Symposium on Antennas and Propagation & USNC/URSI National Radio Science Meeting, July 2018, pp.1887-1888.
- Wagih, M.; Hilton, G.; Weddell, A.; and Beeby, S.; “Broadband Millimeter-Wave Textile-Based Flexible Rectenna for Wearable Energy Harvesting,” IEEE Transactions on Microwave Theory and Techniques, Vol. 68, Issue 11, November 2020, pp. 4960 – 4972.
- Chandravanshi, S.; Katare, K., and Akhtar, M.; “A Flexible Dual-Band Rectenna With Full Azimuth Coverage,” IEEE Access, February 2021 Vol. 9, pp. 27476 – 27484.
- N. A. Eltresy, O. M. Dardeer, A. Al-Habal, E. Elhariri, A.H. Hassan, A. Khattab, D. N. Elsheakh, S A. Taie, H.Mostafa, H. A. Elsadek, and E. A. Abdallah, “RF Energy Harvesting IoT System for Museum Ambience Control with Deep Learning” Sensors, October, pp.1-23, 2019.
Comments 2: The authors need to involve the necessary design equations for the antenna-- For this comment, the authors need to show the detailed calculation from the provided equations and they must comment how the design parameters dimensions are finalized?
Response: Thank you for this comment and we added a paragraph that explains the initial design of the monopole antenna in section 2.2 as follow:
The effective dielectric constant ( ) could be expressed as by using Eq. (1):
(1)
The resonance frequency of clipart moon shaped radiating patch could be calculated by using Eqs (2) and (3) according to the standard formula [8, 9].
The first resonant frequency fr
(2)
(3)
Where ( ) is the velocity of light, ( ) is resonant of frequency, D1 is the outer diminution of clipart moon shaped ( ) is dielectric constant of substrate; h is the height of the substrate. So, using above equations; ; then the first resonant frequency is 3.6 GHz, the ground plane has curvature and simple clipart moon shaped cut to improve the impedance bandwidth and reduced the resonant frequency to be 3 GHz.
Comments 3: In Figure 3, the role of antenna 3 is not clear in the design evolution as compared to antenna 2. Why authors scientifically adopted antenna 3? The answers authors provided for this question are not at all satisfactory.
Response:
Antenna 2 is monopole with circular slot etched in the radiator modified ground plane as shown in following figure.
Antenna 3 is used same monopole radiator with etched rectangular U slot in the modified ground under the 50 ohm transmission line to improve the impedance bandwidth and matching. As shown in reflection coefficient response.
|
|
|
|
Antenna 2 |
Antenna 3 |
|
(a) (b) Figure 1, Simulated reflection coefficient for design steps 2 and 3 by using two simulators (a) CST and (b) HFSS.
|
|
Comments 4: The answers given by authors for comments 5, and 7 are not at all logical. The authors need to show VNA measurement set up image.
Response: It is indeed true that there is a difference between simulated and measured of S11 and VSWR. The VNA is recalibrated and measured again the two parameters as shown in the following figures. The photo is inserted in figure 5 in revised version of the manuscript.
- (b)
Figure 2, Photo of the fabricated antenna setup (a) Reflection coefficient |S11| and (b) VSWR.
- (b)
Figure 3, Photo of the fabricated antenna setup (a) Reflection coefficient |S11| and (b) VSWR.
(c) (d)
Figure 4, Manufactured prototype of the developed UWB antenna (a) front part, (b) back part, the Simulated and measured of the proposed antenna of the UWB antenna structure (c) |S11| and (d) VSWR.
Comments 5: The figures must be straight line for better clarity
Response: Thank you for your comments, about using solid and dash line in the figures to differentiate the response when any reader printed it as black and white. If it printed as solid line with different colours, reader could not able to differentiate between them.
Comments 6: The radiation pattern data seems quite surprising, So much close agreement with simulation is not possible?????
Response:
A detailed explanation of the process of antenna fabrication and the techniques used for experimental measurements are given below.
Rhode & Schwartz ZVA67 with frequency rang 10 MHz–67 GHz is used to measure |S11| over the frequency range 2–24 GHz. Then the custom-made anechoic chamber as shown in figure 6 in ERI Lab., is used to measure gain and normalized radiation pattern measurement. The reference home-made horn antenna boresight is directed towards the antenna under test (AUT) as shown in Figure 6 and it is not wideband over the whole operating frequencies. The AUT is mounted on a rotator that can be used to rotate the antenna about its axis in its horizontal and elevation planes. There are many error could be attributed with the radiation pattern measurements as coaxial cable used need to assemble. Moreover, at high frequency the resulting diffuse field is created and can be absorbed as well as the hard reflecting wave. We added paragraphs that explained the difference of radiation pattern.
“There are difference between measured and simulated radiation pattern especially at high frequencies. These could be attributed due to the custom-made anechoic chamber as shown in figure 6 in ERI Lab. Moreover, the reference home-made horn antenna it is not wideband over the whole operating frequencies. As well as the absorber material not used for high frequency it is limited to 5 GHz.”

Round 3
Reviewer 3 Report
Intentionally the authors have not shown the clear separate image of VNA snapshot.
From figure 5, clearly it can be seen that the authors have manipulated the data and presented the same in comparison to simulated results.
No response letter has been attached. other remarks are not answered.